# Insight into the Anticancer Activity of Copper(II) 5-Methylenetrimethylammonium-Thiosemicarbazonates and Their Interaction with Organic Cation Transporters

**DOI:** 10.3390/biom10091213

**Published:** 2020-08-20

**Authors:** Miljan N. M. Milunović, Oleg Palamarciuc, Angela Sirbu, Sergiu Shova, Dan Dumitrescu, Dana Dvoranová, Peter Rapta, Tatsiana V. Petrasheuskaya, Eva A. Enyedy, Gabriella Spengler, Marija Ilic, Harald H. Sitte, Gert Lubec, Vladimir B. Arion

**Affiliations:** 1Institute of Inorganic Chemistry, Faculty of Chemistry, University of Vienna, Währinger Strasse 42, A-1090 Vienna, Austria; 2Department of Chemistry, Moldova State University, A. Mateevici Street 60, MD-2009 Chisinau, Moldova; palamarciuco@gmail.com (O.P.); sirbuangela@yandex.ru (A.S.); 3Petru Poni Institute of Macromolecular Chemistry, Laboratory of Inorganic Polymers, Aleea Grigore Ghica Voda, Nr. 41A, 700487 Iasi, Romania; shova@icmpp.ro; 4Elettra—Sincrotrone Trieste S.C.p.A, Strada Statale 14—km 163,5 in AREA Science Park, 34149 Basovizza, Trieste, Italy; dan.dumitrescu@gmail.com; 5Institute of Physical Chemistry and Chemical Physics, Faculty of Chemical and Food Technology, Slovak University of Technology in Bratislava, Radlinského 9, SK-81237 Bratislava, Slovakia; dana.dvoranova@stuba.sk (D.D.); peter.rapta@stuba.sk (P.R.); 6Department of Inorganic and Analytical Chemistry, Interdisciplinary Excellence Centre, University of Szeged, Dóm tér 7, H-6720 Szeged, Hungary; petrashevtanya@chem.u-szeged.hu (T.V.P.); enyedy@chem.u-szeged.hu (E.A.E.); 7MTA-SZTE Lendület Functional Metal Complexes Research Group, University of Szeged, Dóm tér 7, H-6720 Szeged, Hungary; spengler.gabriella@med.u-szeged.hu; 8Department of Medical Microbiology and Immunobiology, University of Szeged, Dóm tér 10, H-6720 Szeged, Hungary; 9Department of Pharmaceutical Chemistry, Faculty of Life Sciences, University of Vienna, A-1090 Vienna, Austria; marija.ii.ilic@gmail.com; 10Institute of Pharmacology, Centre for Physiology and Pharmacology, Medical University of Vienna, A-1090 Vienna, Austria; harald.sitte@meduniwien.ac.at; 11Neuroproteomics, Paracelsus Private Medical University, 5020 Salzburg, Austria; gert.lubec@lubeclab.com

**Keywords:** thiosemicarbazones, copper(II) complexes, cytotoxicity, organic cation transporters (OCT1–3), inhibitors

## Abstract

A series of four water-soluble salicylaldehyde thiosemicarbazones with a positively charged trimethylammonium moiety ([H_2_L^R^]Cl, R = H, Me, Et, Ph) and four copper(II) complexes [Cu(HL^R^)Cl]Cl (**1**–**4**) were synthesised with the aim to study (i) their antiproliferative activity in cancer cells and, (ii) for the first time for thiosemicarbazones, the interaction with membrane transport proteins, specifically organic cation transporters OCT1–3. The compounds were comprehensively characterised by analytical, spectroscopic and X-ray diffraction methods. The highest cytotoxic effect was observed in the neuroblastoma cell line SH-5YSY after 24 h exposure and follows the rank order: **3** > **2** > **4** > **cisplatin** > **1** >> [H_2_L^R^]Cl. The copper(II) complexes showed marked interaction with OCT1–3, comparable to that of well-known OCT inhibitors (decynium 22, prazosin and corticosterone) in the cell-based radiotracer uptake assays. The work paves the way for the development of more potent and selective anticancer drugs and/or OCT inhibitors.

## 1. Introduction

The development of thiosemicarbazones (TSCs) as anticancer drugs has a long history. The first compound was tested in vivo in the 1950s, attracting the interest of researchers [1]. Since then, a large number of TSCs with antiproliferative activity has been reported, but only some of them reached clinical trials [2].The academic interest in TSCs has been recently rekindled, when two new compounds, namely COTI-2 ((*E*)-N′-(5,6,7,8-tetrahydroquinolin-8-ylidene)-4-(pyridine-2-yl)piperazine-1-carbothiohydrazide) and DpC (di-2-pyridylketone 4-cyclohexyl-4-methyl-3-thiosemicarbazone), entered phase I clinical trials for the treatment of gynaecological malignancies, colorectal, lung, pancreatic cancer and advanced (or resistant) tumours [3,4].TSCs and their metal complexes are intracellularly reactive agents with multi-target features [5,6,7,8,9,10,11,12,13,14], which often exhibit marked cytotoxic effects both in vitro and in vivo, as described elsewhere [1,2,15,16]. The aromatic moiety and metal-binding domain are the basic structural requisites for their pharmacological activity. The substitutions at the aromatic ring and terminal N atom of the thiosemicarbazide fragment offer additional opportunities for tuning their electronic and steric properties, design and synthesis of more potent drug candidates (Figure 1) [17]. 

From the bio-physico-chemical point of view, the biologically active TSCs must possess: (i) the ability to cross the cell membrane—in order to reach sufficiently high concentrations in the cells, (ii) the proper pharmacological properties to interact with vital intracellular enzymes (or other targets) and induce apoptosis, (iii) redox potentials of their metal complexes falling in a biologically accessible window, that would make them susceptible to intracellular oxidants and reductants, thus allowing for redox cycling between two oxidation states (e.g., Fe^2+^ ↔ Fe^3+^, Cu^+^ ↔ Cu^2+^) and generation of reactive oxygen species (ROS), which can be involved in the mechanism of their anticancer activity [4,11,18]. TSCs have generally limited water solubility, while the well-balanced lipo-, hydrophilic character of a drug candidate molecule is an important feature. The drug should be lipophilic enough to facilitate passive transport through the cell membrane, a property that is often related to enhanced cytotoxicity [11,19,20]. Therefore, fine-tuning the hydrophilic/lipophilic character of the TSC in order to achieve an optimal aqueous solubility and high cytotoxicity is a challenge in the development of more effective anticancer drugs.

Recently, we reported a number of TSC-hybrids with good aqueous solubility by attachment of polar organic molecules, such as L-(D)-proline [21], homoproline [22], amino-esters [23], morpholine [24] to the aromatic moiety of TSCs. Simultaneously, the increase of lipophilic character of the TSCs by structural modification at the terminal amine group of thiosemicarbazide moiety had a beneficial effect on cytotoxicity of the hybrid proligands and their copper(II) complexes [21,23,24,25]. Lipophilic TSCs can enter the cell via passive diffusion, while hydrophilic and charged TSCs have to be transported across the plasma membrane. The bidirectional passage of the molecules through the plasma or intracellular membranes is mediated by polyspecific machinery of large (40–200 kDa) transporter proteins [26,27,28]. Transport of the organic cations through the cell membrane is mediated by three subtypes of the solute carrier (SLC22) family: organic cation transporters (OCTs) namely, OCT1, OCT2 and OCT3 [26]. The OCTs are present in the human body mostly in epithelial cells, neurons, hepatocytes, muscle, and glial cells [26]. Recently, it was reported that OCT3 might be associated with the mitochondria membrane [29]. Moreover, the expression of OCTs was detected in several human cancer cell lines [30]. In colon carcinoma, hOCT1 is expressed at relatively high levels, while human neuroblastoma (SH-SY5Y) and human glioblastoma (HTZ-146) cells demonstrated a significant OCT2 expression [31]. Interestingly, in some human colon adenocarcinoma cell lines, mRNA of all three OCTs was found [32].

The substrates of OCTs include endogenous compounds, e.g., choline, creatinine, monoamine neurotransmitters, and a variety of xenobiotics, such as tetraethylammonium (TEA; a prototypic organic cation), 1-methyl-4-phenylpyridinium (MPP^+^; a neurotoxin), and clinically used drugs, such as metformin (antidiabetic), cimetidine and amantadine (anticancer), which are positively charged at physiological pH [30,32,33]. The most specific substrate for functional studies of OCTs is MPP^+^, exhibiting high maximal uptake rates [26,27,34]. For hOCT2 and hOCT3, in addition to cation influx, cation efflux has also been demonstrated [26,27].

Clinically approved anticancer drugs interact with OCTs. Oxaliplatin is transported by OCT1–3 [26,27,30], OCT2 modulates the uptake of cisplatin, bleomycin and doxorubicin, while OCT1 is involved in the uptake of an anticancer platinum drug (Bamet-UD2) and daunorubicin [35]. In addition to desired drug effects, uptake transporters were recently reported to mediate their side effects [36].

Nevertheless, data about TSCs and their interaction with OCTs have not been reported so far. Passive diffusion through the cell membrane has been reported for lipophilic TSCs [37]. Quite recently, it was suggested that the transport of TSCs and their copper(II) complexes, which are positively charged at physiological pH, may involve active-carrier influx or protein-dependent efflux processes [24,38]. In addition, it was reported that membrane transporters are involved in TSC accumulation in the cell [39]. Moreover, positively charged TSCs might be trapped in the acidic lysosome and bound to copper(II), whereupon generation of ROS the rupture of lysosomes occurs leading ultimately to cell death [40].

The molecules with a trimethylammonium group ([-NMe_3_]^+^) were supposed to use organic cation transporters (OCTs) to act as Trojan horses for metal cations [41,42]. The presence of this cationic group can also increase the bioavailability and enhance the antiproliferative effect by electrostatic interaction with DNA polyanion [41,42].

Herein, we report on the synthesis and characterisation of four salicylaldehyde thiosemicarbazone (STSC) proligands (**[H_2_L^R^]Cl**, R = H, Me, Et, Ph) and their copper(II) complexes (**1**–**4**) shown in Figure 2, in order to elucidate their solution speciation, electrochemical properties, cytotoxic effect and underlying mechanism, as well as the interaction with OCTs. The antiproliferative activity of the compounds was investigated against doxorubicin-sensitive (Colo 205), multidrug-resistant (Colo 320, overexpressing ABCB1 (MDR1)-LRP) human colon adenocarcinoma, neuroblastoma (SH-SY5Y) cell lines and the non-cancerous human embryonal lung fibroblast cell line (MRC-5). The interaction with membrane proteins was studied in HEK cells overexpressing organic cation transporters (OCT1, OCT2 and OCT3) by evaluating their ability to inhibit [^3^H]-MPP^+^ uptake. The structure–activity relationships (SARs) were also discussed, both with respect to cytotoxicity and OCT inhibition.

## 2. Experimental Section

### 2.1. Chemicals

All reagents were purchased from Sigma-Aldrich (Schnelldorf, Germany), Acros Organics (Geel, Belgium) or Alfa Aesar (Kandel, Germany), and used without further purification. The chloride salt of 5-(methylenetrimethylammonium) salicylaldehyde was prepared according to the published procedure [43]. KCl, KOH, HCl was obtained from Reanal (Hungary), KH_2_PO_4_, Na_2_HPO_4_, GSH, AA and 4-(2-hydroxyethyl)-1-piperazineethanesulfonic acid (HEPES) were purchased from Sigma-Aldrich. Copper(II) stock solution was prepared by the dissolution of anhydrous CuCl_2_ in water and its concentration was determined by complexometry with EDTA. Elemental analysis of proligands **[H_2_L^R^]Cl** (R = H, Me, Et, Ph) and complexes **1**–**4** was performed on a Carlo Erba microanalyser at the Microanalytical Laboratory of the University of Vienna. Electrospray ionisation (ESI) mass spectra were measured on a Bruker Esquire 3000 instrument (Bruker Daltonic, Bremen, Germany) at Mass Spectrometry Centre of the Faculty of Chemistry of the University of Vienna. Infrared spectra were recorded on Perkin-Elmer FT–IR 2000 instrument (Watham, MA, USA)(4000–400 cm^−1^) using ATR unit or Bruker Vertex 70 FT–IR spectrometer. UV/Vis spectra were acquired on Agilent 8453 UV/Vis spectrometer (Agilent Technologies, Waldbron, Germany). All samples for NMR measurements were prepared by dissolving the compounds in [D_6_]DMSO. ^1^H, ^13^C, COSY, HSQC, HMBC NMR spectra were acquired on a Bruker Avance III 500 MHz FT NMR spectrometer at NMR spectrometry Centre of the Faculty of Chemistry of the University of Vienna. ^1^H and ^13^C NMR shifts were referred relative to residual solvent signal. The splitting of proton resonances in the ^1^H NMR spectra are defined as singlet (s), doublet (d), doublet of doublets (dd), triplet (t) and multiplet (m).

### 2.2. Synthesis of the Proligands

**General method**. The chloride salt of 5-(methylenetrimethylammonium) salicylaldehyde (10 mmol) was dissolved in MeOH (20 mL) and added to the solution of the corresponding thiosemicarbazide (10 mmol) in MeOH/H_2_O = 1:3 (20 mL). The reaction mixture was heated at 65 °C for 20 min. After partial evaporation of the solvent at room temperature under reduced pressure the yellow crystalline product was filtered off, washed with MeOH (3 mL) and dried in air.
**[H_2_L^H^]Cl·1.4H_2_O** (*E*-isomer). Yield: 85.0%; *E*-isomer ^1^H NMR (500 MHz, [D_6_]DMSO, 25 °C): *δ* = 11.51 (s, 1H; N^2′^H), 10.67 (s, 1H; C^2^-OH), 8.36 (s, 1H; C^11^H), 8.30 (s, 1H; N^3′^-H), 8.10 (s, 1H; C^6^H), 7.81 (s, 1H; N^3′^-H), 7.35 (dd, 1H, *J* = 8.4 Hz, 2.2 Hz; C^4^-H), 7.05 (d, 1H, *J* = 8.4 Hz; C^3^H), 4.41 (s, 2H; C^7^H_2_), 3.01 ppm (m, 9H; 3× C^8–10^H_3_); ^13^C NMR (126 MHz, [D_6_]DMSO, 25 °C): *δ* = 177.8 (C^12^), 157.9 (C^2^), 138.2 (C^11^), 135.0 (C^4^), 130.9 (C^6^), 120.8 (C^1^), 119.0 (C^5^), 116.5 (C^3^), 67.8 (C^7^), 51.6 ppm (C^8–10^); IR (ATR): *ῦ* = 3365, 3322, 3240, 3153, 1593, 1521, 1445, 1363, 1249, 1167, 1087, 968, 870, 844, 822, 751 cm^−1^; MS (ESI): *m*/*z* (%): 267 (25) [*M*]^+^, 208 (100) [*M*-NMe_3_]^+^; elemental analysis calcd (%) for C_12_H_19_ClN_4_OS·1.4H_2_O: C 43.94, H 6.70, N 17.08, S 9.77; found: C 44.26, H 6.55, N 16.88, S 9.98.**[H_2_L^Me^]Cl·H_2_O**. Predominant *E*-isomer (ca. 85%). Yield: 70.3%; ^1^H NMR (500 MHz, [D_6_]DMSO, 25 °C): *δ* = 11.55 (s, 1H; N^2′^-H), 10.61 (s, 1H; C^2^-OH), 8.44 (s, 1H; N^3′^-H), 8.35 (s, 1H; C^11^-H), 8.20 (s, 1H; C^6^-H), 7.35 (dd, 1H, *J* = 8.4 Hz, 2.2 Hz; C^4^-H), 7.05 (d, 1H, *J* = 8.3 Hz; C^3^-H), 4.44 (s, 2H; C^7^H_2_), 3.05–3.00 ppm (m, 12H; 3× C^8–10^H_3_ and C^13^H_3_). ^13^C NMR (126 MHz, [D_6_]DMSO, 25 °C): *δ* = 177.6 (C^12^), 157.8 (C^2^), 137.9 (C^11^), 134.8 (C^4^), 131.4 (C^6^), 120.9 (C^1^), 119.4 (C^5^), 116.5 (C^3^), 67.7 (C^7^), 51.6 (C^8–10^), 30.8 ppm (C^13^). IR (ATR): ῦ = 3285, 3138, 2998, 1613, 1532, 1243, 1083, 1040, 975, 879, 828, 762, 640 cm^−1^; MS (ESI): *m*/*z* (%): 281 (15) [*M*]^+^, 222 (100) [*M*-NMe_3_]^+^; elemental analysis calcd (%) for C_13_H_21_ClN_4_OS·H_2_O: C 46.63, H 6.92, N 16.73, S 9.58; found: C 47.02, H 6.87, N 16.71, S 9.81.**[H_2_L^Et^]Cl**·**0.6H_2_O.** Predominant *E*-isomer (ca. 87%). Yield: 77.1%. ^1^H NMR (500 MHz, [D_6_]DMSO, 25 °C): *δ* = 11.49 (s, 1H; N^1′^-H), 10.66 (s, 1H; OH), 8.47 (s, 1H; N^3′^-H), 8.35 (s, 1H; C^11^-H), 8.22 (s, 1H; C^6^-H), 7.35 (dd, 1H, *J* = 8.4 Hz, 2.1 Hz; C^4^-H), 7.09-7.03 (m, 1H; C^3^-H), 4.47 (s, 2H; C^7^H_2_), 3.62 (dq, 2H, C^13^H_2_), 3.03 (s, 9H; 3× C^8–10^H_3_), 1.16 ppm (t, 3H, *J* = 7.1 Hz; C^14^H_3_); ^13^C NMR (126 MHz, [D_6_]DMSO, 25 °C): *δ* = 176.6 (C^12^), 157.8 (C^2^), 138.1 (C^11^), 134.8 (C^4^), 131.4 (C^6^), 120.8 (C^1^), 119.1 (C^5^), 116.5 (C^3^), 67.4 (C^7^), 51.5 (C^8–10^), 38.3 (C^13^), 14.6 ppm (C^14^); IR (ATR): ῦ = 3355, 3197, 1612, 1531, 1486, 1271, 1222, 1077, 971, 921, 878, 843, 804, 752 cm^−1^; MS (ESI): *m*/*z* (%): 295 (16) [*M*]^+^ 236 (100) [*M*-NMe_3_]^+^; elemental analysis calcd (%) for C_14_H_23_ClN_4_OS·0.6H_2_O: C 49.21, H 7.14, N 16.40, S 9.38; found: C 48.93, H 6.84, N 16.64, S 10.00.**[H_2_L^Ph^]Cl·H_2_O** (*E*-isomer). Yield: 78.4%. ^1^H NMR (500 MHz, [D_6_]DMSO, 25 °C): *δ* = 11.94 (s, 1H; N^2′^-H), 10.81 (s, 1H; C^2^-OH), 9.94 (s, 1H; N^3′^-H), 8.49 (s, 1H; C^11^-H), 8.20 (s, 1H; C^6^-H), 7.64 (d, 2H; C^14^-H and C^18^-H ), 7.43-7.33 (m, 3H; C^4^-H, C^15^-H and C^17^-H), 7.20 (t, 1H, *J* = 7.3 Hz; C^16^-H), 7.11 (d, 1H, *J* = 7.2 Hz; C^3^-H), 4.45 (s, 2H; C^7^H_2_), 3.03 ppm (s, 9H; 3× C^8–10^H_3_); ^13^C NMR (126 MHz, [D_6_]DMSO, 25 °C): *δ* = 175.73 (C^12^), 158.2 (C^2^), 139.2 (C^11^), 138.9 (C^13^), 135.2 (C^4^), 131.3 (C^6^), 128.1 (C^15^ and C^17^), 125.4 (C^16^), 125.2 (C^14^ and C^18^), 120.5 (C^1^), 119.1 (C^5^), 116.5 (C^3^), 67.6 (C^7^), 51.5 ppm (C^8–10^); IR (ATR): ῦ = 3552, 3454, 3277, 2949, 2854, 2715, 1612, 1531, 1500, 1441, 1268, 1203, 1070, 971, 872, 833, 746, 699, 645, 583 cm^−1^; MS (ESI): *m*/*z* (%): 343 (100) [*M*]^+^, 284 (30) [*M*-NMe_3_]^+^; elemental analysis calcd (%) for C_18_H_23_ClN_4_OS·H_2_O: C 54.47, H 6.35, N 14.11, S 8.08; found: C 54.55, H 6.04, N 13.94, S 8.03.

### 2.3. Synthesis of Copper(II) Complexes

**General method**. To the solid mixture of the corresponding proligand **[H_2_L^R^]Cl** (1 equiv) and CuCl_2_·2H_2_O (1 equiv) water (10 mL) was added and the suspension was stirred at 50 °C until a clear solution has been obtained. Afterwards, EtOH (30 mL) was added and the solution was allowed to stand in an open beaker at room temperature. The green crystalline product was filtered off, washed with EtOH (5 mL) and dried in air.
**[Cu(HL^H^)Cl]Cl·2.5H_2_O (1)**. Yield: 71.9%. IR (ATR): ῦ = 3277, 3071, 2815, 2691, 1625, 1534, 1474, 1351, 1178, 967, 875, 825, 727, 677 cm^−1^; UV/Vis (H_2_O): λ_max_ (*ε*) = 623 (125), 370 (11,466), 316 (18,475), 309 nm (18,401 mol^−1^dm^3^cm^−1^); MS (ESI): *m*/*z* (%): 328 (100) [Cu(**L^H^**)]^+^, 268 (80) [Cu(**L^H^**–NMe_3_)]^+^; elemental analysis calcd (%) for C_12_H_18_Cl_2_CuN_4_OS·2.5H_2_O: C 32.33, H 5.20, N 12.56, S 7.20; found: C 32.46, H 4.94, N 12.28, S 7.20.**[Cu(HL^Me^)Cl]Cl·2.5H_2_O (2)**. Yield: 65.7%. IR (ATR): ῦ = 3334, 3235, 3020, 2978, 2885, 2796, 1607, 1528, 1476, 1349, 1175, 1032, 921, 876, 826, 760, 710, 616 cm^−1^; UV/Vis (H_2_O): λ_max_ (*ε*) = 621 (208), 368 (16,980), 316 (24,249), 307 nm (25,137 mol^−1^dm^3^cm^−1^); MS (ESI): *m*/*z* (%): 342 (100) [Cu(**L^Me^**)]^+^, 283 (98) [Cu(**L^Me^**–NMe_3_)]^+^; elemental analysis calcd (%) for C_13_H_20_Cl_2_CuN_4_OS·2.5H_2_O: C 33.95, H 5.48, N 12.18, S 6.97; found: C 33.96, H 5.30, N 11.78, S 6.86. Single crystals suitable for X-ray diffraction analysis were obtained by slow evaporation of EtOH/H_2_O (3:1) solution of **2** at room temperature.**[Cu(HL^Et^)Cl]Cl·2.5H_2_O (3)**. Yield: 66.7%. IR (ATR): ῦ = 3352, 3215, 3024, 2978, 2850, 1586, 1530, 1476, 1352, 1174, 1049, 921, 878, 829, 621 cm^−1^; UV/Vis (H_2_O): λ_max_ (*ε*) = 616 (188), 368 (12,159), 318 (17,433), 308 nm (17,715 mol^−1^dm^3^cm^−1^); MS (ESI): *m*/*z* (%): 356 (100) [Cu(**L^Et^**)]^+^, 297 (90) [Cu(**L^Et^**–NMe_3_)]^+^; elemental analysis calcd (%) for C_14_H_22_Cl_2_CuN_4_OS·2.5H_2_O: C 35.48, H 5.74, N 11.82, S 6.77; found: C 35.67, H 5.41, N 11.60, S 6.97. The X-ray diffraction quality single crystals were obtained by slow evaporation of aqueous solution of **3** at room temperature.**[Cu(HL^Ph^)Cl]Cl·5H_2_O (4)**. Yield: 73.2%; IR (ATR): ῦ = 3437, 3364, 3119, 2955, 2849, 2798, 1584, 1542, 1479, 1443, 1325, 1257, 1176, 970, 923, 874, 827, 758, 640 cm^−1^; UV/Vis (H_2_O): λ_max_ (*ε*) = 601 (115), 382 (16,534), 323 nm (20,281 mol^−1^dm^3^cm^−1^); MS (ESI): *m*/*z* (%): 404 (100) [Cu(**L^Ph^**)]^+^, 345 (80) [Cu(**L^Ph^**–NMe_3_)]^+^; elemental analysis calcd (%) for C_18_H_22_Cl_2_CuN_4_OS·5H_2_O: C 38.13, H 5.69, N 9.88, S 5.66; found: C 38.17, H 5.48, N 9.70, S 5.81. The X-ray diffraction quality single crystals were obtained by slow evaporation of EtOH/H_2_O (3:1) solution of **4** at room temperature.


### 2.4. Spectroscopic Studies (UV/Vis and ^1^H NMR Titrations, Kinetic Measurements and Lipophilicity Determination)

An Agilent Carry 8454 diode array spectrophotometer was used to record the UV/Vis spectra in the interval 200–800 nm. The path length was 1 cm. Proton dissociation constants (p*K*_a_) of the TSC ligands, the overall stability constants of the copper(II) monoligand complexes (log*β*) and the individual spectra of the species in the various protonation states and stoichiometry were calculated by the computer program PSEQUAD [44]. Spectrophotometric titrations were performed on samples containing the proligands at 50 μM concentration by a KOH solution in the presence of 0.1 M KCl at 25.0 ± 0.1 °C in the pH range from 2 to 11.5, and the metal-to-ligand ratio was 1:0, 1:1 and 1:2. An Orion 710A pH-meter equipped with a Metrohm combined electrode (type 6.0234.100) and a Metrohm 665 Dosimat burette were used for the titrations. The electrode system was calibrated to the pH = −log[H+] scale by means of blank titrations (HCl vs. KOH) according to the method suggested by Irving et al [45]. The average water ionisation constant (p*K*_w_) is 13.76 ± 0.05 in water. Argon was also passed over the solutions during the titrations. Measurements were also carried out in the range of ca. 1.0–2.0 by preparing individual samples in which KCl was partially or completely replaced by HCl and pH values (between 1 and 2) were calculated from the strong acid content.

The redox reactions of the copper(II) complexes with GSH and AA were studied at 25.0 ± 0.1 °C on Hewlett Packard 8452A diode array spectrophotometer using a special, tightly closed tandem cuvette (Hellma Tandem Cell, 238-QS). The reactants were separated until the reaction was triggered. Both isolated pockets of the cuvette were completely deoxygenated by bubbling a stream of argon for 10–15 min before mixing the reactants. Spectra were recorded before and then immediately after the mixing, and changes were followed until no further absorbance change was observed. One of the isolated pockets contained the reducing agent and its concentration was 1250 μM and the other contained 25 μM copper(II) complex. The pH of all the solutions was adjusted to 7.40 by 50 mM HEPES buffer and an ionic strength of 0.1 M (KCl) was applied. The stock solutions of the reducing agents and the complexes were freshly prepared every day. During the calculations, the absorbance (*A*) − time (*t*) curves were fitted and analysed at the *λ*_max_ of the complex. (*A*_0_ − *A*_final_) × e^(−*a*×*t*)^ + *A*_final_ equation was used where *A*_0_, *A*_final_ and *a* parameters were refined and accepted at the minimal value of the weighted sum of squared residuals (difference between the measured and calculated absorbance values) at the given wavelength. Then observed rate constants (*k*_obs_) of the redox reaction were obtained from the data points of the simulated absorbance–time curves as the slope of the ln(*A*/*A*_0_) versus *t* plots.

^1^H NMR spectroscopic studies were carried out on a Bruker Avance III HD instrument (Bruker BioSpin GmbH, Rheinstetten, Germany). All spectra were recorded with the WATERGATE water suppression pulse scheme using DSS internal standard.

Distribution coefficients (*D*_7.4_) values were determined by the traditional shake-flask method in *n*-octanol/buffered aqueous solution at pH 7.40 (20 mM phosphate buffer saline (PBS)) at 25.0 ± 0.2 °C as described previously [46,47].

### 2.5. X-ray Crystallography

X-ray diffraction data for **2** and **4** were collected with an Oxford-Diffraction XCALIBUR E CCD diffractometer equipped with graphite-monochromated Mo*K*α radiation. The unit cell determination and data integration were carried out using the CrysAlis package of Oxford Diffraction [48]. Data collection for **3** was performed at the XRD2 structural biology beamline, Sincrotrone Elettra SCpA, Trieste, Italy. The beamline is equipped with Arinax MD2S diffractometer and Pilatus 6M area detector. X-ray radiation is provided by a superconducting wiggler followed by a dual crystal monochromator. Frame integration was performed using the XDS program package [49]. The structures were solved by direct methods using Olex2 software (version 1.3) with the SHELXS structure solution program and refined by full-matrix least-squares on *F*^2^ with SHELXL-2015 [50]. Hydrogen atoms were placed in fixed, idealised positions and refined as rigidly bonded to the corresponding non-hydrogen atoms. Crystal data and details of data collection and structure refinement are provided in Table 1, CCDC-2000723 (**2**), CCDC-2000721 (**3**), CCDC-2000722 (**4**). These data can be obtained free of charge [51] (or from the Cambridge Crystallographic Data Centre, 12 Union Road, Cambridge CB2 1EZ, UK; fax: (+44) 1223-336-033; or deposit@ccdc.ca.ac.uk).

### 2.6. Electrochemical and Spectroelectrochemical Studies

Cyclic voltammetric experiments for **1**–**4** (c_1–4_ = 10^−4^ M) in DMSO (SeccoSolv max. 0.025% H_2_O, Merck) or in MeCN and water using *n*Bu_4_NPF_6_ (puriss quality from Fluka; dried under reduced pressure at 70 °C for 24 h before use, c = 0.1 M) as supporting electrolyte, as well as in deionised water solution at pH 7 (using pH 7 buffer capsules from Sentec) + NaCl (analytical purity, Slavus Ltd.,Bratislava, Slovakia), were performed under argon atmosphere using a three electrode arrangement with glassy-carbon or platinum 1 mm disc working electrodes (from Ionode, Australia), platinum wire as counter electrode, and silver wire as pseudoreference electrode. Ferrocene (from Sigma-Aldrich) served as the internal potential standard for non-aqueous systems, while potassium hexacyanoferrate(II) trihydrate (≥99.95%, Sigma-Aldrich) was used as the internal standard for aqueous solutions. A Heka PG310USB (Lambrecht, Germany) potentiostat with a PotMaster 2.73 software package served for the potential control in voltammetric studies. In situ ultraviolet-visible-near-infrared (UV/Vis/NIR) spectroelectrochemical measurements were performed on a spectrometer (Avantes, Model AvaSpec-2048 × 14-USB2 in the spectroelectrochemical cell kit (AKSTCKIT3) with the Pt-microstructured honeycomb working electrode, purchased from Pine Research Instrumentation (Lyon, France). The cell was positioned in the CUV–UV Cuvette Holder (Ocean Optics, Ostfildern, Germany) connected to the diode-array UV/Vis/NIR spectrometer by optical fibers. UV/Vis/NIR spectra were processed using the AvaSoft 7.7 software package. Halogen and deuterium lamps were used as light sources (Avantes, Model AvaLight-DH-S-BAL).

### 2.7. EPR Spectroscopy, ROS Generation and Antioxidative Activity

The generation of paramagnetic intermediates was monitored by cw-EPR spectroscopy using the EMXplus spectrometer and spin trapping technique. Deionised water was used for preparation of aqueous solutions. The spin trapping agent 5,5-dimethyl-1-pyrroline *N*-oxide (DMPO; Sigma-Aldrich) was distilled prior to the application. The EPR spectra were measured with following experimental parameters: X-band, room temperature, microwave frequency, 9.448 GHz; 100 kHz field modulation amplitude, 2 G; time constant, 10 ms; scan time, 21 s (three scans). Glutathione from Sigma-Aldrich; 6-hydroxy-2,5,7,8-tetramethylchroman-2-carboxylic acid (Trolox, 97%, Sigma-Aldrich); K_2_S_2_O_8_ (p.a. purity, Merck); 2,2′-azinobis [3-ethylbenzothiazoline-6-sulfonic acid] (ABTS; purum, >99%; Fluka) were used as received. Deionised high-purity grade H_2_O, with conductivity of 0.055 mS/cm, was produced by using the TKA H_2_O purification system (Water Purification Systems GmbH, D-Niederelbert). For the ABTS decolourisation assay, the radical cations were pre-formed by the reaction of an aqueous solution of K_2_S_2_O_8_ (3.30 mg) in H_2_O (5 mL) with ABTS (17.2 mg). The resulting bluish-green radical cation (ABTS^•+^) solution was stored overnight in the dark. Before experiment, the solution (1 mL) was diluted into a final volume (60 mL) with H_2_O. The optical spectra were recorded from 1 to 40 min in 1 cm quartz UV cuvette after mixing the sample water/DMSO solution with ABTS^•+^ in mixed solvent water/DMSO (20% (*v*/*v*)).

### 2.8. Cytotoxic Activity

#### 2.8.1. Cell Lines

Human doxorubicin-sensitive Colo 205 (ATCC-CCL-222) and multidrug resistant Colo 320 (overexpressing ABCB1 (MDR1)-LRP (ATCC-CCL-220.1)) colonic adenocarcinoma cell lines were purchased from LGC Promochem, Teddington, UK. The cells were cultured in RPMI 1640 medium supplemented with 10% heat-inactivated fetal bovine serum, 2 mM L-glutamine, 1 mM Na-pyruvate and 100 mM Hepes. The cell lines were incubated at 37 °C, in a 5% CO_2_, 95% air atmosphere. The semi-adherent human colon cancer cells were detached with Trypsin-Versene (EDTA) solution for 5 min at 37 °C.

SH-SY5Y (ATCC^®^ CRL-2266™) neuroblastoma cell line was purchased from LGC Promochem, Teddington, UK. The cell line was cultured in Eagle’s Minimal Essential Medium (EMEM, containing 4.5 g/L glucose) supplemented with a non-essential amino acid mixture, a selection of vitamins and 10% heat-inactivated foetal bovine serum. The cell line was incubated at 37 °C, in a 5% CO_2_, 95% air atmosphere.

MRC-5 human embryonal lung fibroblast cell line (ATCC CCL-171) was purchased from LGC Promochem, Teddington, UK. The cell line was cultured in Eagle’s Minimal Essential Medium (EMEM, containing 4.5 g/L glucose) supplemented with a non-essential amino acid mixture, a selection of vitamins and 10% heat-inactivated foetal bovine serum. The cell line was incubated at 37 °C, in a 5% CO_2_, 95% air atmosphere.

#### 2.8.2. Cytotoxicity Assay

In the study non-cancerous human embryonic lung fibroblast MRC-5, human colonic adenocarcinoma cell line (doxorubicin-sensitive Colo 205, multidrug resistant Colo 320) and neuroblastoma cell line SH-SY5Y were used to determine the effect of compounds on cell growth. The effects of increasing concentrations of compounds on cell growth were tested in 96-well flat-bottomed microtiter plates. The compounds were diluted in a volume of 100 μL of medium.

The adherent human embryonal lung fibroblast and SH-SY5Y neuroblastoma cells were cultured in 96-well flat-bottomed microtiter plates, using EMEM supplemented with 10% heat-inactivated foetal bovine serum. The density of the cells was adjusted to 2 × 10^4^ cells in 100 μL per well, the cells were seeded for 24 h at 37 °C, 5% CO_2_, then the medium was removed from the plates containing the cells, and the dilutions of compounds previously made in a separate plate were added to the cells in 200 μL.

In case of the colonic adenocarcinoma cells, the two-fold serial dilutions of compounds were prepared in 100 μL of RPMI 1640, horizontally. The semi-adherent colonic adenocarcinoma cells were treated with Trypsin–Versene (EDTA) solution. The cells were adjusted to a density of 2 × 10^4^ cells in 100 μL of RPMI 1640 medium, and were added to each well, with the exception of the medium control wells. The final volume of the wells containing compounds and cells was 200 μL.

The culture plates were incubated at 37 °C for 24 h; at the end of the incubation period, 20 μL of MTT (thiazolyl blue tetrazolium bromide, Sigma) solution (from a stock solution of 5 mg/mL) were added to each well. After incubation at 37 °C for 4 h, 100 μL of sodium dodecyl sulfate (SDS) (Sigma) solution (10% in 0.01 M HCI) were added to each well and the plates were further incubated at 37 °C overnight. Cell growth was determined by measuring the optical density (OD) at 540/630 nm with Multiscan EX ELISA reader (Thermo Labsystems, Cheshire, WA, USA). The inhibition of the cell growth (expressed as IC_50_: inhibitory concentration that reduces by 50% the growth of the cells exposed to the tested compounds) was determined from the sigmoid curve where 100 − ((OD_sample_ − OD_medium control_)/(OD_cell control_ − OD_medium control_)) × 100 values were plotted against the logarithm of compound concentrations. Curves were fitted by GraphPad Prism software [52] using the sigmoidal dose-response model (comparing variable and fixed slopes).

#### 2.8.3. Antiproliferation Assay

The method is similar to the one described in the assay for cytotoxic effect. In this assay testing the inhibition of cell proliferation, 6 × 10^3^ cells were distributed in 100 μL of medium with the exception of the medium control wells. The culture plates were incubated at 37 °C for 72 h and after the incubation time, the plates were stained with MTT according to the previous experimental protocol described for the cytotoxicity assay.

##### [^3^H]-MPP^+^ Uptake Inhibition Assay

HEK–293 cells (LGC Standards GmbH, Germany) stably expressing human isoforms of wild type organic cation transporters 1–3 (OCT1–3) were cultured in Dulbecco’s modified Eagle’s medium (DMEM; Sigma-Aldrich) with high glucose (4.5 g/L) and L-glutamine (584 mg/L) supplemented with 10% foetal calf serum (FCS; Biowest) and penicillin/streptomycin mixture (50 mg/L; Sigma/Aldrich) at 37 °C in a humidified atmosphere (5% CO_2_). The selection reagent was Geneticin (150 U/mL; Merck, Germany) for all three cell lines. Cells were seeded onto poly-D-lysine (0.05 mg/mL; Sigma-Aldrich) precoated clear bottom Corning 96 well white assay plate (Sigma-Aldrich) at the density of 4 × 10^4^ cells per well 24 h prior to the experiment.

The assay was performed as previously described with slight modification [53,54]. Briefly, on the day of the experiment, cells were washed once with 100 µL Krebs-HEPES buffer (KHB; 10 mM HEPES, 120 mM NaCl, 3 mM KCl, 2 mM CaCl_2_·2H_2_O, 2 mM MgCl_2_·6H_2_O, 5 mM D-(+)- glucose, pH 7.3) at room temperature. Cells were preincubated for 10 min in 50 µL of RT KHB containing increasing concentrations of tested compounds previously dissolved in dimethyl sulfoxide (DMSO; final concentration in working solution was 1%). After that, the preincubation solution was quickly removed and replaced with 50 µL of uptake solution containing the aforementioned components with the addition of 50 nM [^3^H]-1-methyl-4-phenylpyridinium ([^3^H]-MPP^+^; 60 Ci/mmol; American Radiolabeled Chemicals, Saint Louis, MO, USA). After the 10 min uptake at RT, the transport was stopped by washing cells with 100 µL of ice-cold KHB and adding 200 µL of Ultima GoldTM XR scintillation cocktail (PerkinElmer, MA, USA). The plates were shaken, placed into Wallac 1450 MicroBeta TriLux Liquid Scintillation Counter and Lumi (GMI, Ramsey, MN, USA) and the released radioactivity was measured for 3 min. Nonspecific [^3^H]-MPP^+^ uptake was measured in the presence of 100 µM decynium-22 and the data were plotted as a percentage of uptake in the presence of 1% DMSO from which the value of the nonspecific uptake was subtracted. Nonlinear regression analysis (GraphPad Prism version 5.0, GraphPad Software, San Diego, CA, USA) was used for calculation of IC_50_ values.

## 3. Results and Discussion

### 3.1. Synthesis and Characterisation of Proligands and Copper(II) Complexes

Two structural modifications of the STSC scaffold were performed: (i) *para*-substitution at the phenol ring and (ii) the introduction of aliphatic and aromatic substituents at the terminal NH_2_ group of the thiosemicarbazide fragment. Both types of substitutions are known to enhance the anticancer potency of TSC derivatives [24,55]. The proligands **[H_2_L^R^]Cl** with R = H, Me, Et and Ph were prepared by the condensation reaction of the chloride salt of 5-(methylenetrimethylammonium) salicylaldehyde [43,56,57].with the corresponding substituted thiosemicarbazide (Appendix A). The starting aldehyde salt was obtained by the reaction of 5-chloromethylsalicylaldehyde with trimethylamine in tetrahydrofuran (THF) by following a literature protocol. The copper(II) complexes **1**–**4** were synthesised by reactions of CuCl_2_·2H_2_O with the corresponding thiosemicarbazone in aqueous solution, and crystallised upon the addition of ethanol. All compounds were isolated in a solid state as chloride salts and characterised by elemental analysis, electrospray ionisation (ESI) mass spectrometry, ^1^H and ^13^C NMR, IR spectroscopy and single-crystal X-ray diffraction (SC-XRD). One and two dimensional NMR spectra (Appendix A) were in agreement with expected structures for **[H_2_L^R^]Cl** of *C*_1_ molecular symmetry enabling for the assignment of all ^1^H and ^13^C resonances (Figure 2). In addition, the spectra measured in [D_6_]DMSO indicated the presence of only an *E* isomer (**[H_2_L^H^]Cl** and **[H_2_L^Ph^]Cl**) or a mixture of two isomers (*E* and *Z*) with the first as favoured isomer (**[H_2_L^Me^]Cl** (85%) and **[H_2_L^Et^]Cl** (87%)). The isomerism is solvent dependent. The aqueous behaviour will be also discussed (vide infra), even though the *E*/*Z* isomerism does not affect their pharmacological profile [58].

The ESI mass spectra of the proligands **[H_2_L^R^]Cl** recorded in positive ion mode showed two main peaks. The most abundant peak was attributed to [**H_2_L^R^**–NMe_3_]^+^, while the second to [**H_2_L^R^**]^+^ ion. Similarly, the copper(II) complexes showed peaks, attributed to [Cu**L^R^**–NMe_3_)]^+^ and [Cu(**L^R^**)]^+^ ions. IR spectra revealed characteristic bands for C=S (1593–1625 cm^−1^) and C=N (804–829 cm^−1^) bonds in both proligands and copper(II) complexes. In UV–visible (UV/Vis) spectra of copper(II) complexes **1**–**4** (Appendix A), d–d transition bands were observed with maxima between 601 and 623 nm (ε = 115–208 mol^−1^dm^3^cm^−1^). The coordination geometry around copper(II) in **2**–**4**, as well as the protonation state of the TSC ligands was further established by SC-XRD. 

### 3.2. Single Crystal X-ray Diffraction Analysis

The results of SC-XRD analysis for **2**–**4** are shown in Figure 3 and Section 2.5. The bond distances and bond angles are quoted in Appendix A. According to this study, two complexes contain singly deprotonated ligands and have an ionic crystal structure, which is built up from the complex cations **[Cu(HL^Me^)Cl]^+^** (for **2**) and **[Cu(HL^Ph^)Cl]^+^** (for **4**) and one chloride counteranion. Compound **3** re-crystallised in water revealed a molecular structure consisting of **[Cu(L^Et^)Cl]** neutral entities, in which the thiosemicarbazone ligand is doubly deprotonated (at the phenolic OH group and the hydrazinic N^2′^-nitrogen). There are two chemically identical but crystallographically independent complex cations in the asymmetric unit of **2**. Note that the crystals of **2**, **3** and **4** contain 2.5, 3.2, and 5.0 molecules of co-crystallised water per complex unit, respectively. In all complexes copper(II) atom adopts a slightly distorted square-planar coordination geometry, provided by a tridentate mono- or dianionic ONS thiosemicarbazone ligand and one chlorido co-ligand (Figure 3). 

The performed characterisation (^1^H and ^13^C NMR, IR, ESI-MS, SC-XRD) of the prepared proligands and complexes provided evidence for the composition and structure of the compounds in the solid state and in solutions of organic solvents. However, the knowledge on their behaviour in aqueous solution along with redox properties is highly important for the understanding of their pharmacological potential.

### 3.3. A Comparative Solution Equilibrium Study on Proligands and Their Copper(II) Complexes 

#### 3.3.1. Proton Dissociation Processes and Lipophilicity of the Proligands

In order to exhibit their anticancer effect, the drug candidates dissolved in the aqueous biological fluids need to be transported through the cell membrane(s) and then activated by intracellular oxidants or reductants. Therefore, we studied the solution speciation of our compounds in aqueous solution in the broad pH range from 1.0 to 11.5 (including the physiological pH), redox properties using cyclic voltammetry and the direct effect of reducing agents (such as glutathione (GSH) and ascorbic acid (AA)). Investigation of the solution speciation is necessary to elucidate protonation/deprotonation and complex formation/dissociation equilibria. Based on this information the most abundant species at physiological pH can be suggested. 

The proton dissociation processes of the proligands **[H_2_L^R^]Cl** were studied primarily by UV/Vis spectrophotometric titrations in aqueous solution in the pH range 1.0–11.5. Representative UV/Vis spectra for **[H_2_L^H^]Cl** in Figure 4a show characteristic pH-dependent changes of the overlapping π→π* and n→π* transition bands originated mainly from the azomethine chromophore (λ_max_ ~ 301 nm) and the phenolic moiety (λ_max_ ~ 327 nm). These proligands contain two dissociable protons, namely the phenolic OH and the hydrazinic-N^2′^H. Only one deprotonation step could be identified in the studied pH range based on the appearance of the isosbestic points at 230, 280 and 347 nm. Therefore, one p*K*_a_ value was determined for each ligand precursor by the deconvolution of the spectra (Table 2) and is attributed to the deprotonation of the phenolic OH group. (Notably, p*K*_a_ for the hydrazinic-N^2′^H moiety, which is expected to be higher than that of the phenolic-OH [59], could not be determined in the studied pH range as its deprotonation takes place at pH > 11.5 where the measurement of the pH becomes uncertain due to the alkaline error of the glass electrode.) The p*K*_a_ value of the phenolic OH moiety of salycilaldehyde TSC is 8.88 determined in pure water [59], thus the studied proligands have more than 1 order of magnitude lower proton dissociation constants than salycilaldehyde TSC due to the electron-withdrawing effect of the trimethylammonium moiety. 

The *N*-terminally monomethylated and monoethylated derivatives have similar p*K*_a_ values to that of the unsubstituted proligand, while the phenyl group at this position results in somewhat stronger acidity. Based on the determined data, concentration distribution curves computed with the p*K*_a_ value(s) (Figure 4b shows distribution curves for **[H_2_L^H^]Cl**) revealed that these originally positively charged compounds are partly deprotonated at physiological pH: 42–55% of the proligand is in the neutral HL^(+/−)^ form in case of **[H_2_L^R^]Cl** (R = H, Me, Et), while 70% in case of **[H_2_L^Ph^]Cl**. The neutral form HL^(+/−)^ is actually zwitterionic, due to the presence of trimethylammonium [-NMe_3_]^+^ and phenolate [Ph-O]^−^ moieties resulting in excellent water solubility.

The lipophilicity of the proligands at pH 7.4 expressed as distribution coefficients (log*D*_7.4_ in Table 2) was determined by the traditional shake flask method by UV/Vis spectrophotometry (Appendix A). The determined log*D*_7.4_ values reveal that the introduction of the methyl, ethyl and/or phenyl group increases the lipophilicity in the following rank order: **[H_2_L^H^]Cl** < **[H_2_L^Me^]Cl** < **[H_2_L^Et^]Cl** < **[H_2_L^Ph^]Cl**, as expected. These compounds are much more hydrophilic than the unsubstituted reference compound, STSC (log*D*_7.4_: +1.74) [59].

As *E/Z* isomerism with respect to the C=N double bond is well documented for TSCs [60], to further investigate the proton dissociation processes and the formation of these isomers ^1^H NMR titration was performed for **[H_2_L^H^]Cl** (Figure 5). Two sets of signals were registered in the whole pH range studied, indicating that the isomers are in slow interconversion processes with regard to the NMR time scale (t_1/2(obs)_ > ~ 1 ms) in this medium. However, the extent of peak separation was dependent on the type of protons. In some cases, the peaks were strongly overlapping (e.g., CH=N; CH_2_; CH_3_). Based on the integrated signals, the molar fraction of the major isomer is ~ 70% at pH 1, and is increased up to a constant value of ~ 85% at pH > 4. The identification of the isomers was mainly based on the chemical shifts of the N^2^H proton of the hydrazinic moiety, and the p*K*_a_ values determined for the isomers (Figure 5) based on the pH-dependence of the chemical shifts (Appendix A). The N^2′^H group is fairly sensitive to the isomer identity and based on literature data [61,62,63], the major species was characterised as the *E* isomer by the N^2′^H resonance at 10.02 ppm at pH ~ 1 which shifted down-field to 10.09 ppm with increasing pH along with the deprotonation of the phenolic OH. At the same time, the N^2′^H peak of the minor, *Z* isomer (at 11.08 ppm) could be seen only in the acidic pH range, most likely due to the hydrogen bond formation between this N^2′^H proton donor and the phenolato group as a proton acceptor (see the suggested formula for the HL form of the minor isomer in Figure 5). On the other hand, the p*K*_a_ calculated for the minor isomer is ca. half logarithm unit lower (6.95 vs. 7.51) than that of the other isomer, which is in accord with the formation of this intramolecular hydrogen bond that makes the deprotonation of the phenolic OH group easier. Notably, ^1^H NMR spectra of the same proligand in [D_6_]DMSO did not provide evidence for isomerism (or the resonance lines collapsed to a single peak as a result of the rapid interconversion of the isomers). Similar behaviour was reported for STSC in 30% [D_6_]DMSO/H_2_O [59].

#### 3.3.2. Solution Stability of Copper(II) Complexes and Their Reduction by GSH

Solution speciation of copper(II) complexes of STSC was described in detail in our previous work [59].However, the measurements were performed in 30% (*w*/*w*) DMSO/H_2_O solvent mixture because of the limited aqueous solubility of STSC. Formation of exclusively monoligand complexes was found, in which the ligand coordinates in a tridentate mode via (O,N,S) donor set. In the acidic pH range, the non-coordinating hydrazinic nitrogen remains protonated. The complex, in which the ligand is coordinated as dianion via deprotonation of the phenolic function and the hydrazinic N^2′^H group was found in the pH range 6–9. It is worth noting that in that complex the negative charge was localised on the sulfur atom instead of the nitrogen due to the thione-thiol tautomeric equilibrium. In addition, the formation of a mixed hydroxido species with (O^−^, S, N^−^)(OH^−^) coordination mode in the basic pH range was documented [59]. By looking at the structure of the two proligands (STSC and **[H_2_L^R^]Cl**) a similar binding pattern is assumed in case of the complexes **1**–**4** studied in this work, even though different solution stability can be envisioned due to the presence of the [-NMe_3_]^+^ moiety, different *N*-terminal substitution and the type of solvent. DMSO is able to coordinate to copper(II) weakly, therefore, somewhat higher copper(II) complex stability is expected in the neat aqueous solution [64].

UV/Vis spectra of the copper(II)–TSC systems were recorded at various metal-to-proligand ratios (see the showcase spectra for Cu(II)–**[H_2_L^H^]Cl** system at Cu:L = 1:1 ratio in Figure 6a). Overall stability constants were calculated for [Cu(HL)]^2+^, [CuL]^+^ and [CuLH_−1_] species (Table 2). Notably, the coordination of the chlorido co-ligand was confirmed in a solid state. However, in solution, the coordination is quite weak, so the binding of the chlorido ligand is negligible. As the protonated form of the proligand was formulated as [H_2_L]^+^, in the [Cu(HL)]^2+^ the ligand in its neutral zwitterionic form coordinates to copper(II) most likely as an (O^−^,S,N) donor. [CuL]^+^ complex is formed by the deprotonation of the hydrazinic nitrogen, and [CuLH_−1_] is a mixed hydroxido [CuL(OH)] species. The p*K*_a_ values of the complexes (Table 2) indicate that the deprotonation of the hydrazinic nitrogen takes place in the pH range 3–5. A considerably lower p*K*_a_ [Cu(HL)]^2+^ for the phenyl derivative was calculated in a more acidic pH range. The formation of the mixed hydroxido complexes is found in the pH range 9–11 in all cases.

The concentration distribution curves computed for the Cu(II)–**[H_2_L^H^]Cl** system show (Figure 6b) that the [CuL]^+^ complex predominates in a wide pH range including the physiological pH. To compare the stability of this type of complexes, a derived stability constant was calculated taking into account the different ligand basicities (see log*K*_derived_[CuL]^+^ values in Table 2). pCu (the negative decadic logarithm of the concentration of the free copper(II) ions) values were also computed for comparison at pH 7.4 (Table 2). Both the log*K*_derived_ [CuL]^+^ and pCu values reveal the following copper(II) binding ability of the proligands: **[H_2_L^Me^]Cl** ~ **[H_2_L^Et^]Cl** < **[H_2_L^H^]Cl** << **[H_2_L^Ph^]Cl**. Thus, the *N*-terminal phenyl substitution increased the copper(II) complex stability, most probably the conjugated electron system of the phenyl group can contribute to the stabilization of the metal-chelate complex. It is worth pointing out that the computed pCu values reflect significantly high stability of the copper(II) complexes at pH 7.4, since <0.1% decomposition of the complexes is estimated at 10 μM concentration.

The log*D*_7.4_ values of the complexes (Table 2) indicate that they are more hydrophilic than the corresponding proligands (Appendix A). The effect of substituent R on the lipophilicity follows the same order as for the proligands. 

The anticancer activity of the copper(II) complexes of TSCs is often related to their reduction by cellular thiols such as GSH[11,62]. GSH is able to reduce the copper(II)–TSC complex to copper(I) species, while re-oxidation with oxygen can generate reactive oxygen species (ROS). The copper(II) complexes of Dp44mT and DpC were reported to be potentially involved in ROS production in lysosomes via redox cycling leading to lysosomal membrane permeabilisation and ultimately to apoptosis [65]. Therefore, we investigated the direct reduction of the copper(II) complexes with GSH spectrophotometrically under the strictly anaerobic condition at pH 7.4. The spectral changes were monitored in the wavelength range of 300–500 nm in the presence of large excess reducing agent (50 equiv.), since in this range, only the spectral changes characteristic to the absorption of the metal complex and the TSC proligand are visible. In addition, ascorbic acid, an abundant low molecular mass reducing agent in the blood, was also tested but the reaction was very slow suggesting that these copper(II) complexes cannot be reduced efficiently by this reducing agent. On the contrary, significant spectral changes were detected with GSH as Figure 7a shows for complex **2** (and Appendix A for **1**). The first recorded spectrum after mixing the reactants showed several shifts of the absorbance bands most likely due to the formation of a ternary complex with GSH as reported for other TSC complexes [64,66,67]. Then a significant decrease of the absorbance was observed at *λ*_max_ = 376 nm, while the absorbance value of the free proligand (*λ*_max_ ~ 329 nm) was increased, presumably due to decomposition of the generated copper(I) complex, which is unstable. By bubbling O_2_ into the solution almost complete regeneration of the copper(II) complexes was observed (see, e.g., Appendix A for **1**), suggesting a quite reversible redox transformation. The other copper(II) complexes showed a similar behaviour.

Copper(I) has a strong preference for tetrahedral coordination geometry [68,69] even though linear [70] and 3-coordinate geometry has also been documented [71,72]. The potentially tridentate thiosemicarbazones, like HL^R^ studied in this work, are unable to accommodate tetrahedral copper(I) [68]. Therefore, a strong rearrangement of the coordination environment occurs in order to meet the copper(I) requirements for one of the named geometries. The potentially tridentate thiosemicarbazones can act as mono- or bidentate in copper(I) complexes, but thione or thiol sulfur as a soft Lewis base always coordinates to copper(I) as a soft Lewis acid in accordance with the hard and soft acids and bases (HSAB) principle. X-ray diffraction study of copper(I) complex with STSC revealed a hexanuclear copper(I) cluster [Cu_6_(L^1^H)_6_]·6DMF·Et_2_O consisting of two associated six-membered rings, each in a chair conformation. Phenyl group insertion at N4-atom resulted in the same type of copper(I) cluster [71]. Rapid and quantitative oxidation of dimethylformamide solutions of distorted tetrahedral copper(I) complexes with bis(thiosemicarbazones) back to square-planar copper(II) complexes was confirmed by UV/Vis spectroscopy and SC-XRD [68].

However, in the case of **4**, which complex was found to be somewhat more stable than **1**–**3**, a slower reaction was observed (see absorbance changes in Figure 7b). In order to obtain comparable data, the recorded absorbance–time curves were further analysed mainly at the *λ*_max_ of the complex. The observed rate constants (*k*_obs_) were calculated (Table 2) as a semi-quantitative description of the reaction kinetics. These *k*_obs_ values represent well the findings described above.

Copper(II) chelation was reported to enhance the antiproliferative efficacy of TSCs against cancer cells, even though it is not a sufficient factor for displaying the cytotoxicity. In addition, TSC complexes should be able to induce redox cycling [11]. Redox cycling between two oxidation states (Cu^2+^↔Cu^+^) is another important feature affecting their anticancer activity [4,11]. During this process, following Fenton-like chemistry, high levels of ROS, e.g., HO^●^ and/or O_2_^●−^ can be formed leading to cell death [4,11,62,66,73]. Redox cycling with the production of ROS was also reported for other essential bioavailable metals, e.g., iron [18]. Another important peculiarity is that this process should occur within the biologically accessible window of potentials (−0.4 to +0.8 V vs. NHE), because the most promising TSC drug candidates showed redox activity in this range of potentials [11,73].

The observed redox activity of **1**–**4** when reacted with biological reductant GSH and reoxidation with O_2_, which appeared to be reversible, prompted us to investigate their redox properties and evaluate the ability to generate ROS under biologically relevant conditions by cyclic voltammetry (CV) and spectroelectrochemistry.

### 3.4. Cyclic Voltammetry and Spectroelectrochemistry

The electrochemical behaviour of **1**–**4** was investigated by cyclic voltammetry and UV/Vis/NIR spectroelectrochemistry in water, DMSO and acetonitrile (MeCN). A characteristic spectroelectrochemical response in aqueous solutions at pH 7 by using phosphate buffer is shown for **1** in Figure 8. A strongly shifted reoxidation peak (see inset in Figure 8b) upon reverse scan is in agreement with significant rearrangement of the coordination environment around copper(I) into the preferred square-planar geometry of copper(II). These findings are in an agreement with the results reported for similar copper(II) complexes [55]. Additionally, it seems that the reduced copper(I) form is less soluble since over time in the region of the first reduction peak a continuous decrease of all optical bands was observed. Precipitation and adsorption of the reduced species on the electrode surface in the thin layer cell appear to occur. However, upon reoxidation, a full recovery of the initial optical bands is observed (see last red traces in Figure 8b). Therefore, a large peak-to-peak separation in the corresponding cyclic voltammogram can be attributed to significant structural differences between the copper(II) and copper(I) redox states. The redox behaviour observed in the region of the first reduction step can be explained by the electrochemical square scheme [74,75,76,77].

A similar effect was observed for **2** and **3** in MeCN/0.1 M *n*Bu_4_NPF_6_ (Appendix A). The chemical reversibility in line with the electrochemical square scheme was further confirmed by standard CV by using a glassy carbon disc electrode at a scan rate of 100 mV s^−1^ for **1** and **3** in phosphate buffer with 0.1 M KCl (pH 7) (Appendix A). The first reduction process clearly occurs in the biologically accessible window of potentials (−0.4 V to +0.8 V vs. NHE or −1.04 V to +0.16 V vs. Fc/Fc^+^) as shown in Appendix A. The potentials were recalculated vs. NHE by using K_4_[Fe(CN)_6_] as internal standard (see the blue trace in Appendix A). When applying two consecutive voltammetric scans going to the cathodic part, an irreversible voltammetric response was obtained (see the black traces in Appendix A), wherein the second voltammetric scan a strong decrease of the cathodic peak was observed (see dashed black trace in Appendix A). However, when going to the anodic part, a new strongly shifted reoxidation peak emerged (see peak marked with an asterisk and red trace in Appendix A), which cannot be attributed to the oxidation of the initial Cu(I) complex (see blue line in Appendix A). In addition, when the redox cycling in the cathodic part was expanded to the strongly shifted reoxidation peak, in the second voltammetric scan, the peak height of the cathodic wave was almost the same as in the first scan (see dashed black trace in Appendix A). More reversible electrochemical behaviour and enhanced solubility of initial as well as reduced forms of **1**–**4** were observed in DMSO as illustrated for **2** (Appendix A). The quasi-reversible cathodic peak was found at *E*_pc_ = −1.1 V vs. Fc^+^/Fc both at glassy carbon (Appendix A) and platinum working electrodes (Appendix A), at a scan rate of 100 mV s^−1^. In the corresponding UV/Vis spectroelectrochemical experiment for **2** a new absorption band at *λ*_max_ = 330 nm upon cathodic reduction at the first cathodic wave appeared (Appendix A). The appearance of this band is even better seen for **3** in MeCN in the cathodic part (Appendix A). These findings correspond well to the results obtained spectrophotometrically, where a new absorption band at *λ*_max_ ~ 330 nm appeared when **1** or **2** was allowed to react with excess GSH (as the reducing agent) in aqueous solution at pH 7.4 or pH 7, respectively (Figure 7, Appendix A). In agreement with reoxidation of the reduced Cu(I) species by O_2_ after the reduction of **1** or **2** with GSH, the spectroelectrochemical measurements of complexes in water confirmed reversible redox processes upon redox cycling. Note that in contrast to **1**–**4**, the corresponding proligands did not exhibit any redox activity in the region of potentials from −1.4 to 0.2 V vs. Fc^+^/Fc (not shown).

The anodic oxidation of the initial complexes in water, acetonitrile and DMSO solutions takes place at relatively high anodic potentials of ca. 0.9 V vs. NHE or ca. 0.4 V vs. Fc^+^/Fc (see the red trace in Appendix A). The first irreversible intense oxidation peak indicates intricate multi-electron irreversible processes in the anodic part. Upon anodic oxidation of **3** in MeCN at the first oxidation peak, a new absorption band at 294 nm arises with a simultaneous decrease of the initial band at 395 nm via an isosbestic point at 318 nm (Appendix A). Note that this process is electrochemically and chemically irreversible under the conditions used. No recovery of the initial absorption band was monitored upon the reverse scan. 

### 3.5. Electron Paramagnetic Resonance Spectroscopy, ROS Generation and Antioxidant Activity in Cell-Free Media

Although the production of ROS by redox reactions has been recently challenged [78], the generation of ROS via thiol-mediated reduction of copper(II) to copper(I) has been assumed as the major mechanism of anticancer activity of many copper(II) TSC complexes [62]. The presence of reducing agents such as biologically relevant GSH is crucial for the redox reactivity of copper(II) TSC complexes [62].

To support the results of spectrophotometric studies, that copper(II) is reduced to copper(I) in the presence of GSH, EPR spectroscopy was also used, as the reduction leads to the EPR silent d^10^ copper(I) species. The reduction process was monitored for **1** and **3** before and after the addition of excess GSH to their aqueous buffer solution (pH 7). A strong decrease of characteristic EPR signal with S = ½ for d^9^ copper(II) state was observed due to the formation of diamagnetic (S = 0) cuprous ions (see red traces in Figure 9a for **1** and Figure 9b for **3**).

To ascertain the ability of the copper(II) complexes to generate ROS in the presence of GSH, the free-radical spin-trapping agent 5,5-dimethyl-1-pyrroline N-oxide (DMPO) was used. The system **1**/DMPO/H_2_O_2_ in aqueous solution was monitored via the EPR spin trapping technique [62] in the absence and presence of GSH (Figure 9c). The reduced state of the complex **1** (formed upon addition of GSH) generates a substantial amount of hydroxyl radicals (^●^OH) in agreement with the EPR spectrum (see red trace in Figure 9c) which showed a signal attributed to hydroxyl radical spin adduct (^●^DMPO-OH). Only a very small amount of ^●^DMPO-OH was detected for the same system in the absence of GSH (see blue trace in Figure 9c). There was no radical formation found for the corresponding proligand **[H_2_L^H^]Cl**. 

In contrast to the prooxidative activity of investigated copper(II) complexes (i.e., ROS generation) confirmed by EPR spin trapping experiments, the antioxidant activity was observed for the corresponding proligands as illustrated for **[H_2_L^H^]Cl** (Appendix A). All proligands contain a phenolic moiety, and, therefore, they can behave as potential antioxidants. 2,2′-Azino-bis(3-ethylbenzothiazoline-6-sulfonic acid radical cation (ABTS^•+^) represents a paramagnetic species, stable at room temperature, usually applied as an oxidant to characterise the antioxidant activity of various systems [79]. UV/Vis spectra of ABTS^•+^ in solvent mixture water/DMSO (20% (*v*/*v*)) are shown for **[H_2_L^H^]Cl** in Appendix A). The decrease in the absorbance at 735 nm was monitored for various initial concentrations in the range of 0–83 μM (Appendix A) and Trolox equivalent antioxidant capacity (TEAC) was determined. The TEAC value of 0.74 indicates similar antioxidant effect as for the standard Trolox (TEAC = 1) (Appendix A). This corresponds well to the antiproliferative effect of the proligands discussed below where negligible activity was observed. Analogous experiments performed for **1** revealed negligible antioxidant activity even at high concentration (Appendix A).

TSCs are excellent chelators for transition metal ions. The metal-binding site (ONS or NNS donor atom sets) in the TSC backbone is an important prerequisite for the development of anticancer drugs, as derivatives without such binding domains showed low or even lack of activity [11]. From the anticancer therapy point of view, the complex formation of TSC with copper(II) is intriguing, as cancers cells rely upon higher intracellular levels of copper relative to healthy cells, to promote angiogenesis, tumour growth and metastasis [80,81,82]. The cytotoxic effect of the most potent TSCs, for example, Dp44mT [40,83] and NSC-319726 [84] in cancer cells is potentiated by chelation to copper(II). Analogous behaviour was also reported for proline and morpholine substituted TSCs hybrids [21,24]. Therefore, we further studied the cytotoxicity of new TSCs and their copper(II) complexes.

### 3.6. Cytotoxicity

The antiproliferative activity of the proligands **[H_2_L^R^]Cl** and copper(II) complexes **1**–**4** against neuroblastoma (SH-SY5Y) cells, doxorubicin-sensitive (Colo205), multidrug-resistant (Colo 320/MDR-LRP) colon adenocarcinoma and non-cancerous embryonal lung fibroblast (MRC-5) was investigated by the colorimetric MTT assay with exposure times of 24 and 72 h. The IC_50_ values for proligands, copper(II) complexes and cisplatin (as control) are shown in Table 3.

The choice of cancer cell lines for antiproliferative activity assays was dictated by the fact that human neuroblastoma (SH-SY5Y) cells demonstrated significant expression of OCT2, while in some colon carcinoma cell lines expression of OCT1 is noticed [31]. Additionally, the uptake of anticancer drug doxorubicin is modulated by OCT2 [35]. Altogether, the OCTs might be considered as potential targets for the development of anticancer agents.

The **[H_2_L^Et^]Cl** and **[H_2_L^Ph^]Cl** derivatives showed moderate cytotoxicity in SH-SY5Y cells after an incubation time of 72 h with IC_50_ of 49.86 μM and 31.46 μM, respectively. The other proligands did not show any significant cytotoxic effects. The chelation to copper(II) increases the antiproliferative effect (up to 11-fold) for proligands **[H_2_L^Et^]Cl** and **[H_2_L^Ph^]Cl** in all cancer cell lines, while for **[H_2_L^H^]Cl** and **[H_2_L^Me^]Cl** an enhancement of cytotoxicity by a factor of ca. 2 and ca. 7, respectively, was found only for SH-SY5Y cell line. The highest cytotoxic potency was observed in SH-SY5Y for complex **3** with IC_50_ values 10.34 μM and 8.93 μM upon exposure time of 24 and 72h, respectively. The same compound in Colo205 and SH-SY5Y cell lines revealed higher activity compared to cisplatin by a factor of ~ 2.5 at 24 h exposure time. Complex **4** exhibited a similar cytotoxic effect to **3** in Colo205 cell line, while it was less potent in Colo320 and SH-SY5Y cell lines. Complex **2** was most potent in the neuroblastoma cell line (IC_50_ = 23.35 μM) with cytotoxicity comparable to cisplatin (IC_50_ = 26.03 μM) and to **3** (IC_50_ = 10.34) at 24 h exposure time, while in other cell lines exhibited lower activity compared to **3** and **4**.

All copper(II) complexes displayed a drop of their activity after increasing the exposure time to 72 h, except for **1–3** in SH-SY5Y, where only a slight increase in the activity was observed. The best cytotoxic/antiproliferative effect of the studied compounds was observed in the neuroblastoma (SH-SY5Y) cell line following the rank order: **3** > **2** > **4** > **cisplatin** > **1** >> **[H_2_L^R^]Cl** according to their IC_50_ values for 24 h exposure time. However, the pattern was dramatically changed after 72 h exposure: **cisplatin** >> **3** > **2** > **[H_2_L^Ph^]Cl** > **1** > **[H_2_L^Et^]Cl** > **4** > **[H_2_L^H^]Cl ~ [H_2_L^Me^]Cl**.

Relying on the selectivity factor (SF) values, defined as the ratio between IC_50_ values of the compounds in noncancerous cell lines and IC_50_ values in cancerous cell lines, the highest selectivity was found for **3** (SF ~ 6) and for **2** (SF ~ 4) in the case of neuroblastoma cells (SH-SY5Y) when the exposure time was 24 h.

### 3.7. Interaction with OCT1, 2 and 3

About 40% of the commonly prescribed drugs exist as organic cations at physiological pH [85]. Membrane transport proteins OCT1–3 are responsible for the influx of positively charged xenobiotics to the cell. OCT1 and OCT2 share about 70% of the amino acid sequence, whereas OCT3 is approximately 50% identical to OCT1 and OCT2 [86].

Even though many compounds have been shown to inhibit or modulate the transport activity of the OCTs, not all of them are transportable substrates. For several compounds, which are substrates for OCTs, the transport has been directly demonstrated[33,87]. A variety of cations, e.g., decynium-22 (D-22) or disprocynium, neutral compounds such as corticosterone and β-estradiol inhibit OCTs, but are not transported themselves [26,27]. The key determinants for substrate/inhibitor binding to OCTs are positive charge and hydrophobic mass of the molecule. In general, bulkier and more hydrophobic cations are often potent inhibitors, but not substrates for the OCTs [87].

At the same time OCT1, OCT2 and OCT3 demonstrate their specificities for substrates and inhibitors. The affinity of the transported substrate and non-transported inhibitor for individual OCTs overlap broadly [26,27]. MPP^+^ is an often-used model cation that is transported by OCT1-3 as it exhibits high maximal uptake rates in all three cation transporters [26,27]. Thus, the inhibition of MPP^+^ uptake represents a useful pharmacological tool for the screening of the ability of a compound to interact with OCTs. Some well-known inhibitors, such as D-22, corticosterone, progesterone, o-methylisoprenaline, prazosin and phenoxybenzamine were tested for their affinity for OCTs by exploiting the [^3^H]-MPP^+^ uptake assay [32,88].

Solution speciation studies (vide supra) established that the proligands **[H_2_L^R^]Cl** are mainly present in cationic [H_2_L]^+^ or zwitterionic form [HL]^+/−^, while copper(II) complexes **1**–**4** as cations at physiological pH. Therefore, their interaction with human OCT1–3 transporters was investigated.

The proligands **[H_2_L^R^]Cl** and **1**–**4** were examined for inhibition of OCT-mediated uptake of [^3^H]-MPP^+^ in HEK293 cells stably expressing OCT1, OCT2 and OCT3. The inhibition curves and calculated IC_50_ values are shown in Figure 10 and Table 4.

Copper(II) complexes **1**–**4** inhibited [^3^H]-MPP^+^-uptake in OCT1, OCT2 and OCT3 with different potency. Complex **4** exhibited the highest potency toward all three OCTs with IC_50_ values in the submicromolar range (0.25–0.62 μM, Table 4). These values are comparable with the IC_50_ values of the high-affinity inhibitors: decynium-22, prazosin and corticosterone shown in Table 4 [33]. Complex **3** was the most potent inhibitor of OCT3 (IC_50_ = 1.34 μM), while less potent by factors 3 and 25 for OCT2 and OCT1, respectively.

The complex **2** was less potent than **3** in competition with [^3^H]-MPP^+^ for all OCTs, namely three-fold less potent for OCT1 and 2-fold less potent for both OCT2 and OCT3 (Figure 10 and Table 4). Complex **1** was the most selective in competition with [^3^H]-MPP^+^ for OCT2 (IC_50_ = 8.5 μM) and for OCT3 (IC_50_ = 6.6 μM), while substantially less selective towards OCT1 (IC_50_ = 286.6 μM). Taking together, the ability of the potential drug candidates **1**–**4** to inhibit the [^3^H]-MPP^+^ cell uptake via OCT1, OCT2 and OCT3 follows the order: **4** >> **3** > **2** > **1**. A certain pattern of inhibition potencies is of note for **1**–**3** regarding specific transporter, i.e., these three complexes inhibited the [^3^H]-MPP^+^ uptake in the following order OCT3 > OCT2 > OCT1, while the pattern was different for complex **4**, namely OCT2 > OCT1 > OCT3.

It should be pointed out that the bulkier groups usually improve the potency of the inhibitors. This observation also holds for **1**–**4** (H < Me < Et < Ph) with respect to all OCTs, while for the proligands, only in the case of OCT3.

The proligands showed substantially higher IC_50_ values compared to **1**–**4**. Moderate potency for OCT1 (IC_50_ = 740.1 μM and IC_50_ = 390.8 μM), while no interaction with OCT2 was observed for **[H_2_L^H^]Cl** and **[H_2_L^Me^]Cl**, respectively. The highest activity of **[H_2_L^H^]Cl** and **[H_2_L^Me^]Cl** was found in the case of OCT3 (IC_50_ = 105.1 μM and IC_50_ = 102.8 μM, respectively). Proligands **[H_2_L^Et^]Cl** and **[H_2_L^Ph^]Cl** showed inhibitory potency for OCT3 that was comparable with that observed for their corresponding complexes **3** and **4** (IC_50_ = 63.1 μM and IC_50_ = 27.8 μM, respectively). The observed IC_50_ values for the inhibition of [^3^H]-MPP^+^ uptake via OCT1 by **[H_2_L^Et^]Cl** and **[H_2_L^Ph^]Cl** were 168.6 µM and 118.9 µM, while the values for the inhibition of [^3^H]-MPP^+^ uptake via OCT2 were 236.2 µM and 370.8 µM, respectively. Interestingly, a different uptake inhibition pattern was observed for proligands (when compared to that for complexes), namely the inhibition uptake follows the order: OCT2 > OCT1 > OCT3 (Appendix A). The poor affinity of the proligands towards OCTs might be explained by their partial abundance as positively charged ions (30–58%) at physiological pH when compared to copper(II) complexes (100%).

## 4. Conclusions

Four thiosemicarbazone proligands and four copper(II) complexes, which contain a trimethylammonium cation attached at salicylaldehyde moiety conferring excellent aqueous solubility, were prepared, isolated in a solid state and characterised by spectroscopic and analytical methods. X-ray diffraction study of the complexes **2**–**4** revealed a slightly distorted square-planar coordination geometry of the copper(II) ion, provided by the (O, N, S) donor atoms of the tridentate TSCs and one chlorido co-ligand.

p*K*_a_ values determined by spectrophotometric titrations revealed that 42–55% of the proligands **[H_2_L^R^]Cl** (R = H, Me, Et) are present in solution in neutral zwitterionic form at physiological pH, while 70% in case of **[H_2_L^Ph^]Cl**.

In contrast, the copper(II) complexes dominate in solution as positively charged **[CuL^R^]^+^** (**[Cu(HL)H_-1_]^+^**) species in a broad pH range including the physiological pH. The binding ability to copper(II) was found to be the highest for the proligand with phenyl moiety and follows the rank order: phenyl >> ethyl > methyl > unsubstituted derivative. This specific difference was found to overlap considerably with the higher potency of **1**–**4** when compared to the proligands to inhibit the [^3^H]-MPP^+^ uptake by HEK cells via OCTs.

The ability of copper(II) complexes to be reduced by GSH was investigated in solution by UV/Vis and EPR spectroscopy. It was disclosed that under the anaerobic conditions at physiological pH, the complexes are reduced to copper(I) species, indeed. The reduction reaction followed by EPR spectroscopy resulted in the formation of EPR silent (S = 0) d^10^ Cu(I) species. These species can be reoxidised in the presence of oxygen to original copper(II) complexes. Thus, copper(II) complexes were found to be redox-active at physiological pH and might react with intracellular reductants. In agreement with these data, the electrochemical and spectroelectrochemical studies of proligands and the copper(II) complexes in DMSO, MeCN and aqueous solution, showed that only the complexes underwent a reduction in biological accessible window (–0.4 to +0.8V vs. NHE), while the proligands remained intact. Thus, the reduction is metal-centred, as described for other copper(II) complexes developed as anticancer agents [55].

The ability to generate ROS is important for the development of anticancer agents, as higher levels of ROS lead to cell apoptosis. The copper(II) complexes reported herein produce ROS in the presence of GSH as evidenced by the EPR spin trap technique in cell-free media. In contrast, the corresponding proligands do not generate ROS; however, they show significant antioxidant activity, unlike **1–4**.

The antiproliferative activity of the proligands and the complexes was investigated in neuroblastoma (SH-SY5Y) cells, doxorubicin-sensitive (Colo205), multidrug-resistant (Colo320) colon adenocarcinoma and non-cancerous embryonal lung fibroblast (MRC-5) by the colorimetric MTT assay with the exposure times 24 and 72 h. The lead compound **3** was the most effective in SH-SY5Y and Colo205 cancer cell lines and by a factor of ~2.5 more potent than control compound-cisplatin when the exposure time was 24 h. The cytotoxic efficacy in SH-SY5Y cell line follows the rank order: **3** > **2** > **4** > **cisplatin** > **1** >> **[H_2_L^1–4^]Cl**. The drop of antiproliferative activity was characteristic for all compounds in all cancer cell lines after an exposure time of 72 h, except for **1**–**3** in SH-SY5Y, where slight enhancement of the activity was observed. The superior anticancer activity of the copper(II) complexes is likely due to their suitable electrochemical reduction potentials and the ability to produce deadly ROS levels in the presence of biological reductants.

The complex formation with copper(II) changes the pharmacological profile of **[H_2_L^R^]Cl**. This is reflected not only in the much higher antiproliferative activity of **1**–**4** in cancer cell lines but also in the favourable effect of metal coordination on the proligand ability to inhibit the [^3^H]-MPP^+^ uptake by HEK cells via organic cation transporters. The inhibition ability of **1**–**4** was further increased by the introduction of bulkier groups at the terminal N atom of thiosemicarbazide fragment. Complex **4** exhibited remarkable inhibitory potency in submicromolar range (IC_50_ values 0.2–0.6 μM), close to those of well-known OCT inhibitors D-22, prazosin and corticosterone. The complexes **1**–**3** inhibited the [^3^H]-MPP^+^ uptake by HEK cells via OCT2 and OCT3 with IC_50_ up to 10 μM, while via OCT1, the inhibitory activity was lower. The complex **3** (with IC_50_ = 1.34 μM) was the most potent for OCT3. Altogether, complexes **1**–**3** were less potent than **4**, but more selective to specific transporter. The following selectivity was established: OCT2 > OCT1 > OCT3 for **4** and OCT3 > OCT2 > OCT1 for **1**–**3**.

It is also worth noting that the most stable (according to pCu value) and the most lipophilic complex **4** (according to log*D*_7.4_ values) at physiological pH exhibited the highest affinity to OCT1–3 in submicromolar range of IC_50_ values. Apart from the more lipophilic character and overall positive charge of the species at physiological pH, the planarity of the complex may play an additional role in the interaction with OCTs. From the structural point of view, the complexes possess a more extended conjugated framework when compared to proligands.

Finally, the copper(II) complexes, particularly complex **3**, showed the highest antiproliferative potency in SH-5YSY cell line, accompanied with a marked affinity for OCT2 and the ability to generate ROS, suggesting that interaction with OCTs might be at least in part responsible for the anticancer activity of this lead compound. Taken together, the OCTs could be considered as possible targets in the future development of TSCs as anticancer agents. Moreover, the selectivity of cytostatic drugs for cancer cells may be enhanced by co-administration with a drug that inhibits the uptake of the cytostatic drug into normal cells but does not interfere with the uptake of the respective cancer cells. The compounds reported herein represent a sound basis for the further development of anticancer drugs and/or OCT inhibitors.

## Figures and Tables

**Figure 1 biomolecules-10-01213-f001:**
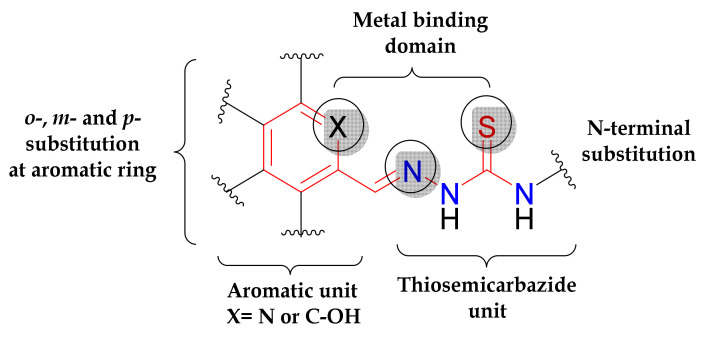
Basic structural features of thiosemicarbazones determining their biological activity.

**Figure 2 biomolecules-10-01213-f002:**
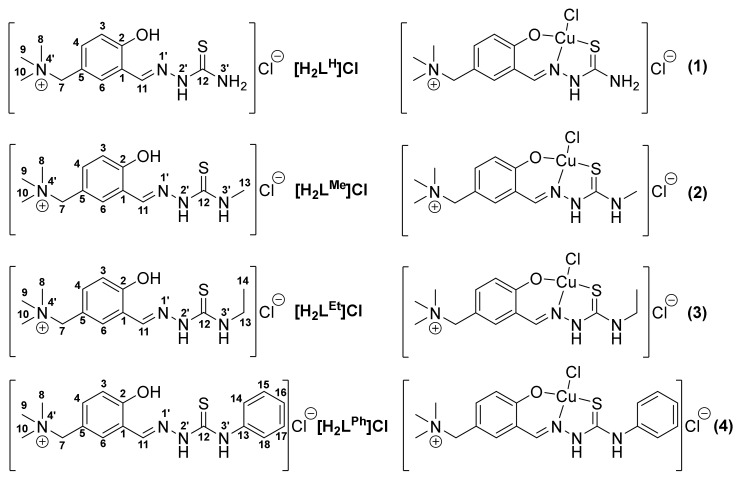
The line drawings of salicylaldehyde thiosemicarbazone (STSC) analogues **[H_2_L^R^]Cl** and their copper(II) complexes **1**–**4**. The C atoms (1–18) and N atoms (1′–4′) labelling in the proligands is used for NMR resonances assignment.

**Figure 3 biomolecules-10-01213-f003:**
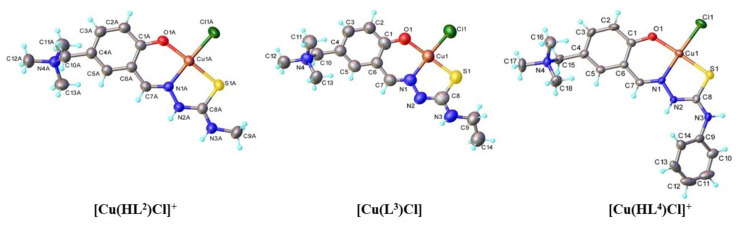
ORTEP (Oak Ridge Thermal-Ellipsoid Plot Program) view of the complexes **[Cu(HL^Me^)Cl]^+^** (**2**) (left), **[Cu(L^Et^)Cl]** (**3**) (middle) and **[Cu(HL^Ph^)Cl]^+^** (**4**) (right) with atom labeling scheme and thermal ellipsoids at 50% probability level. In the case of **2**, only one of the crystallographically independent cations (**A**) is shown. Selected bond distances (Å) and bond angles (deg) for **2**: Cu1–O1 = 1.910(2), Cu1–N1 = 1.960(2), Cu1–S1 = 2.2577(10), Cu1–Cl1 = 2.2397(10); O1–Cu1–N1 = 92.44(10), N1–Cu1–S1 = 85.85(8); for **3**: Cu1–O1 = 1.899(5), Cu1–N1 = 1.979(5), Cu1–S1 = 2.245(2), Cu1–Cl1 = 2.240(2); O1–Cu1–N1 = 92.5(2), N1–Cu1–S1 = 86.51(17); for **4**: Cu1–O1 = 1.908(3), Cu1–N1 = 1.971(3), Cu1–S1 = 2.2562(12), Cu1–Cl1 = 2.2653(11); O1–Cu1–N1 = 91.49(12), N1–Cu1–S1 = 85.96(9). The solvents and counter-anions are omitted for clarity.

**Figure 4 biomolecules-10-01213-f004:**
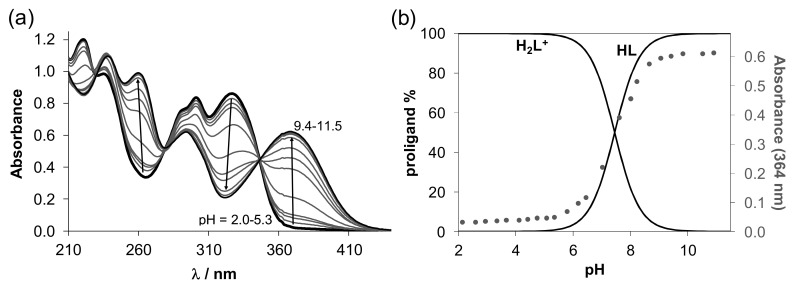
(**a**) UV/Vis absorption spectra of proligand **[H_2_L^H^]Cl** at various pH values, and (**b**) its concentration distribution curves plotted together with the absorbance changes at 364 nm (●) (*c*_L_ = 50 μM; *I* = 0.1 M (KCl); *t* = 25 °C).

**Figure 5 biomolecules-10-01213-f005:**
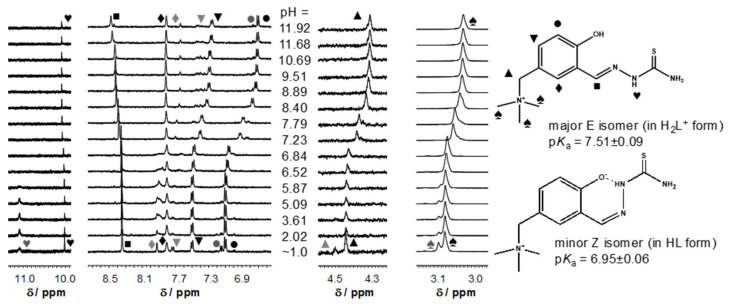
^1^H NMR spectra of the proligand **[H_2_L^H^]Cl** (with different zooming of the selected regions for the better visibility) at various pH values with symbols used for proton resonances assignment in case of the major E isomer (black symbols) and minor Z isomer (grey symbols) and their p*K*_a_ values calculated from the chemical shift changes (*c*_L_ = 200 μM; *I* = 0.1 M (KCl); *t* = 25 °C; 10% (*v*/*v*) D_2_O/H_2_O).

**Figure 6 biomolecules-10-01213-f006:**
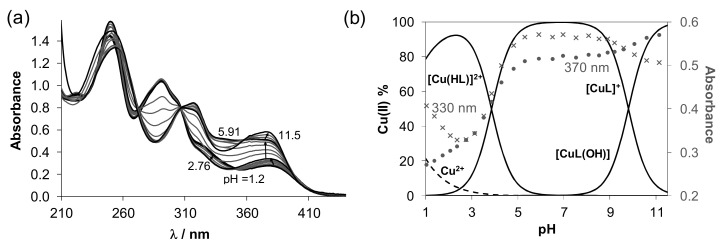
(**a**) UV/Vis absorption spectra recorded for copper(II) complex of **[H_2_L^H^]Cl** (**1**) at various pH values, and (**b**) its concentration distribution curves plotted together with the absorbance changes at 330 (×) and 370 nm (●) (*c*_complex_ = 50 μM; *I* = 0.1 M (KCl); *t* = 25 °C).

**Figure 7 biomolecules-10-01213-f007:**
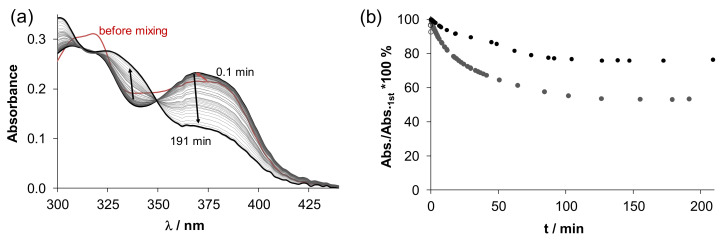
(**a**) Time-dependent UV/Vis absorption spectra of **2** in the presence of 50 equiv. glutathione (GSH) before (red line) and after mixing the reactants (black lines) in a tandem cuvette. (**b**) Absorbance values at 370 nm (grey dots) for **2** and at 376 nm (black dots)for **4** independent of time. Absorbance read from the first spectrum recorded after mixing is considered as 100%, the empty symbols denote the absorbance values measured before mixing (pH = 7.4 (50 mM HEPES); *c*_complex_ = 25 μM; *c*_GSH_ = 1.25 mM; *I* = 0.1 M (KCl); *t* = 25 °C). The measurement was done anaerobically.

**Figure 8 biomolecules-10-01213-f008:**
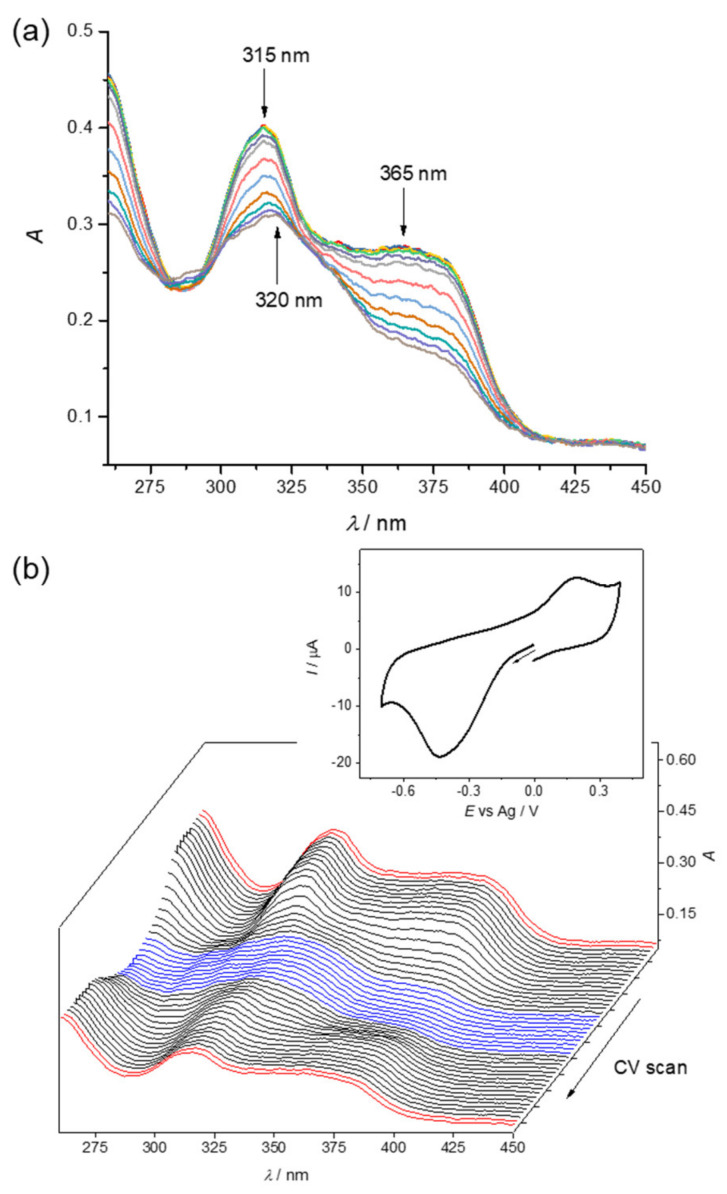
UV/Vis spectra measured upon cathodic reduction of ca. 0.1 mM aqueous solution of **1** at pH 7 at the first reduction peak by using a honeycomb platinum working electrode (scan rate 10 mV s^−1^). (**a**) UV/Vis spectra measured upon cathodic reduction in the forward scan and (**b**) UV/Vis spectra measured during one cyclic voltammetric scan (inset: the corresponding in situ cyclic voltammogram).

**Figure 9 biomolecules-10-01213-f009:**
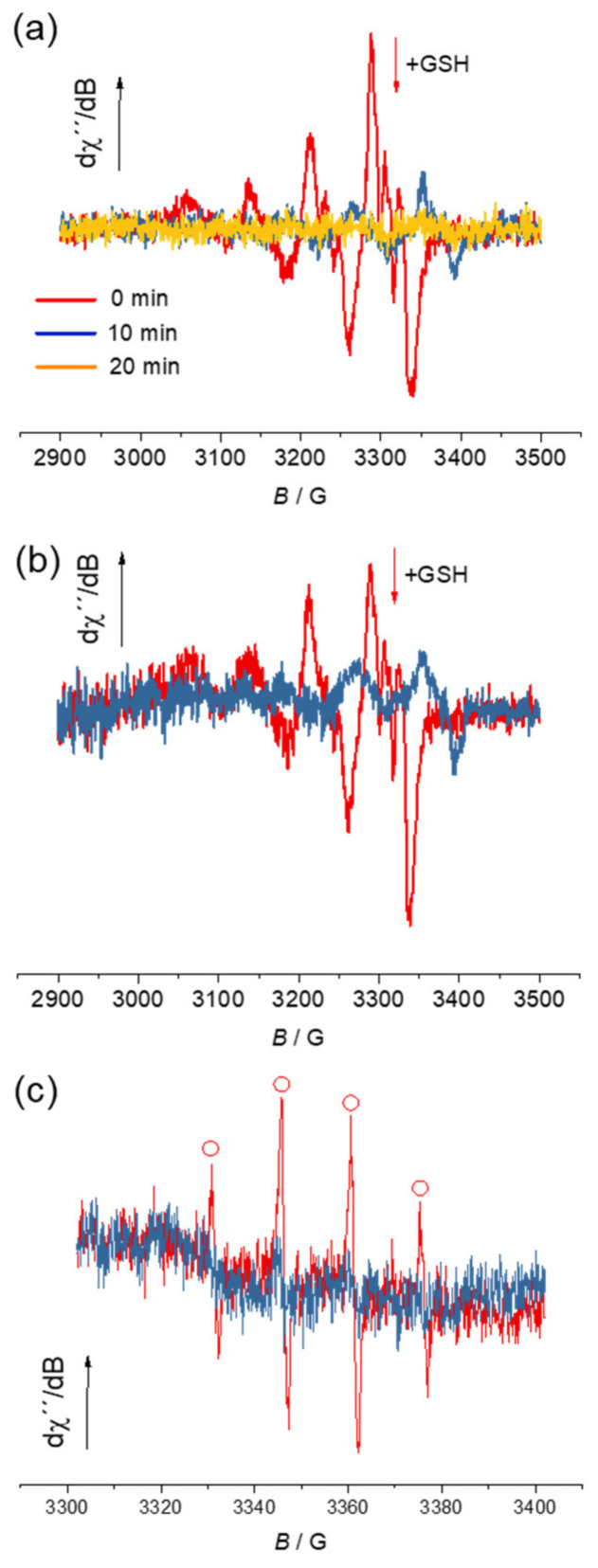
(**a**) EPR spectra of ca. 0.1 mM of **1** in aqueous solution at pH 7 measured before (red trace) and after addition of excess GSH after 10 min (blue trace) and 20 min (dark yellow trace) of the reaction; (**b**) EPR spectra of ca. 0.1 mM of **3** in aqueous solution at pH 7 measured before (red trace) and after addition of excess GSH after 10 min (blue trace) of the reaction; (**c**) EPR spectra of DMPO spin-adducts for **1** in water + DMPO + H_2_O_2_ system measured on air after 2 min of reactions (blue trace) and for **1** in water + GSH + DMPO + H_2_O_2_ system measured on air after 2 min of reactions (red trace): *c*_0_(**1**) = 0.4 mM, *c*_0_(H_2_O_2_) = 0.01 M, *c*_0_(DMPO) = 0.04 M, *c*_0_(GSH) = 1.2 mM.

**Figure 10 biomolecules-10-01213-f010:**
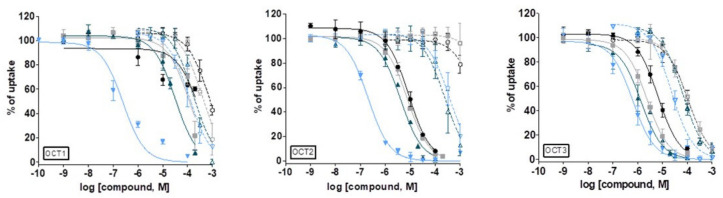
Concentration-response curves of **[H_2_L^R^]Cl** and **1**–**4** for the inhibition of [^3^H]-MPP^+^ uptake via organic cation transporter (OCT) 1–3. HEK cells stably expressing human isoforms of OCT1–3 were incubated with the increasing concentrations of the tested drugs and tritiated substrate, [^3^H]-MPP^+^, for 10 min. Curves were fitted using nonlinear regression and data points are expressed as the mean ± SD of 3–4 experiments performed in triplicates. **[H_2_L^H^]Cl** (○), **[H_2_L^Me^]Cl** (**□**), **[H_2_L^Et^]Cl** (
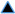
), **[H_2_L^Ph^]Cl** (
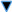
), **1** (●), **2** (■), **3** (▲), **4** (▼).

**Table 1 biomolecules-10-01213-t001:** Crystallographic data for **2**–**4**.

Parameter	2	3	4
Empirical formula	C_26_H_49.81_Cl_4_Cu_2_N_8_O_6.91_S_2_	C_14_H_28.45_ClCuN_4_O_4.22_S	C_18_H_32_Cl_2_CuN_4_O_6_S
Formula weight	918.03	451.51	566.97
Temperature/K	293	100	293
Crystal system	triclinic	monoclinic	monoclinic
Space group	*P*-1	*C*2/*c*	*P*2_1_/*c*
*a*/Å	11.8147(4)	16.951(3)	8.7395(8)
*b*/Å	13.5950(5)	25.050(5)	34.147(2)
*c*/Å	13.7575(5)	9.7040(19)	8.8692(6)
*α/°*	89.099(3)		
*β*/°	65.774(4)	106.39(3)	108.701(9)
*γ/°*	86.133(3)		
*V*/Å^3^	2010.38(14)	3952.9(15)	2507.1(4)
*Z*	2	8	4
*D*_calc_/mg/mm^3^	1.517	1.517	1.502
*μ*/mm^−1^	5.143	1.320	1.208
Crystal size/mm^3^	0.30 × 0.10 × 0.02	0.05 × 0.04 × 0.015	0.01 × 0.01 × 0.01
*θ*_min_,*θ*_max_(°)	6.516 to 133.18	2.94 to 48.586	6.028 to 50.05
Reflections collected	12,762	20,627	16,513
Independent reflections	70,383 [*R*_int_ = 0.0327]	3294 [*R*_int_ = 0.0780]	4421 [*R*_int_ = 0.0718]
Data/restraints/parameters	7038/0/467	3294/0/253	4421/0/292
GOF ^c^	0.984	1.04	1.043
*R*_1_^a^ (*I* > 2σ(*I*))	0.0482	0.0826	0.0584
*wR*_2_^b^ (all data)	0.1395	0.2586	0.1035
Largest diff. peak/hole/e Å^−3^	0.76/−0.84	1.08/−0	0.34/−0.38

^a^*R*_1_ = Σ||*F*_o_| − |*F*_c_||/Σ|*F*_o_|. ^b^
*wR*_2_ = {Σ[*w*(*F*_o_^2^ − *F*_c_^2^)^2^]/Σ[*w*(*F*_o_^2^)^2^]}^1/2^. ^c^ GOF = {Σ[*w*(*F*_o_^2^ − *F*_c_^2^)^2^]/(*n* − *p*)}^1/2^, where *n* is the number of reflections and *p* is the total number of parameters refined.

**Table 2 biomolecules-10-01213-t002:** Proton dissociation constants (p*K*_a_) of the studied proligands; overall stability constants (log*β*), p*K*_a_ and derived constants (log*K*_derived_) of their copper(II) complexes, pCu (=−log[Cu(II)]) values calculated (*I* = 0.1 M (KCl); *t* = 25 °C), distribution coefficients (log*D*_7.4_) of the proligands and copper(II) complexes at pH 7.4, as well as the observed rate constants (*k*_obs_) for the redox reaction of the complexes with GSH (pH = 7.4; c_complex_ = 25 μM; c_GSH_ = 1.25 mM).

	[H_2_L^H^]Cl	[H_2_L^Me^]Cl	[H_2_L^Et^]Cl	[H_2_L^Ph^]Cl
p*K*_a_ (H_2_L^+^)	7.46 ± 0.01	7.54 ± 0.01	7.31 ± 0.01	7.03 ± 0.01
% HL at pH 7.4	47%	42%	55%	70%
log*D*_7.4_ (proligand)	−0.84 ± 0.03	−0.39 ± 0.01	0.06 ± 0.04	0.68 ± 0.04
log*β* [Cu(HL)]^2+ [a]^	12.00 ± 0.01	11.73 ± 0.02	11.66 ± 0.03	11.26 ± 0.02
log*β* [CuL]^+ [b]^	8.14 ± 0.01	7.76 ± 0.02	7.67 ± 0.03	8.55 ± 0.02
log*β* [CuLH_−1_] ^[c]^	−1.66 ± 0.02	−1.48 ± 0.08	−1.65 ± 0.09	−1.31 ± 0.05
p*K*_a_ [Cu(HL)]^2+^	3.86	3.97	3.99	2.71
p*K*_a_ [CuL]^+^	9.80	9.24	9.32	9.86
log*K*_derived_ [CuLH_−1_]^+ [d]^	0.68	0.22	0.36	1.52
pCu ^[e]^	12.21	11.79	11.82	12.80
log*D*_7.4_ (complex)	−1.00 ± 0.01	−0.79 ± 0.01	−0.40 ± 0.01	−0.17 ± 0.01
*k*_obs_ (min^−1^)	0.080 ± 0.008	0.073 ± 0.002	0.058 ± 0.001	0.025 ± 0.005

^[a]^*β* [Cu(HL)]^2+^ = [Cu(HL)]^2+^/[Cu^2+^] × [HL]. ^[b]^
*β* [CuL]^+^ = [CuL]^+^ × [H]^+^/[Cu^2+^] × [HL]. ^[c]^
*β* [CuLH_−1_] = [CuLH_−1_] × [H]^+^ × [H]^+^/[Cu^2+^] × [HL]. [CuLH_−1_] = [CuL(OH)]. ^[d]^ log*K*_derived_ = log*β* [CuL]^+^ − p*K*a (H_2_L^+^) for the equilibrium: Cu^2+^ + H_2_L^+^ [CuL]^+^ + 2H^+^. ^[e]^ pCu = −log[Cu(II)] at pH = 7.4; c_Cu(II)_ = 10 μM; c_L_ = 10 μM.

**Table 3 biomolecules-10-01213-t003:** IC_50_ values after inhibition of cell growth by the proligands **[H_2_L^R^]Cl** and **1**–**4** in human doxorubicin-sensitive (Colo205), multidrug-resistant (Colo320) adenocarcinoma, neuroblastoma (SH-SY5Y) cell lines and non-cancerous human embryonal lung fibroblast (MRC-5).

Compound	Colo205 (IC_50_, μM) ^[a]^		Colo320 (IC_50_, μM)		SH-SY5Y (IC_50_, μM)		MRC-5 (IC_50_, μM)	
	24 h ^[b]^	72 h ^[b]^	24 h	72 h	24 h	72 h	24 h	72 h
**[H_2_L^H^]Cl**	>100 ^[c]^	>100	>100	88.77 ± 5.32	>100	>100	>100	>100
**[H_2_L^Me^]Cl**	>100	>100	>100	74.43 ± 3.85	>100	>100	>100	>100
**[H_2_L^Et^]Cl**	>100	>100	63.83 ± 3.92	53.30 ± 2.44	88.13 ± 9.19	49.86 ± 3.97	>100	>100
**[H_2_L^Ph^]Cl**	>100	>100	>100	95.81 ± 1.58	>100	31.46 ± 2.66	>100	>100
**1**	88.79 ± 4.40	>100	>100	65.38 ± 3.71	45.44 ± 9.1	38.15 ± 1.15	>100	>100
**2**	73.4 ± 2.24	53.34 ± 3.31	>100	80.64 ± 1.57	23.35 ± 3.57	15.35 ± 1.74	>100	51.53 ± 4.67
**3**	27.51 ± 1.68	28.59 ± 2.02	30.47 ± 3.27	35.41 ± 1.87	10.34 ± 2.65	8.93 ± 0.5	57.79 ± 2.65	22.6 ± 1.81
**4**	36.99 ± 3.06	42.81 ± 4.23	45.02 ± 2.49	65.14 ± 2.93	38.07 ± 5.18	58.64 ± 7.18	52.71 ± 6.49	25.64 ± 2.57
**Cisplatin**	68.82 ± 5.08	8.14 ± 1.59	12.69 ± 0.79	2.12 ± 0.14	26.03 ± 2.38	0.65 ± 0.08	55.67 ± 4.06	1.73 ± 0.28

^[a]^ IC_50_ values were calculated as mean values obtained from three independent experiments. IC_50_ values are quoted with their standard deviations (SD). ^[b]^ IC_50_ values were established after the exposure time of 24 and 72 h. ^[c]^ The sign > indicates that the IC_50_ value is not reached in the examined range of concentrations.

**Table 4 biomolecules-10-01213-t004:** [^3^H]-MPP^+^ uptake inhibition by **[H_2_L^1−4^]Cl** and **1**–**4** in HEK OCT1–3 cells.

Compound	OCT1 (IC_50_, μM) ^[a]^	OCT2 (IC_50_, μM) ^[a]^	OCT3 (IC_50_, μM) ^[a]^
**[H_2_L^1^]Cl**	740.2 ± 194.8	>4000	105.1 ± 32.8
**[H_2_L^2^]Cl**	390.8 ± 40.0	>4000	102.8 ± 23.4
**[H_2_L^3^]Cl**	168.6 ± 26.2	236.2 ± 97.1	63.1 ± 8.9
**[H_2_L^4^]Cl**	118.9 ± 50.0	370.8 ± 123.2	27.8 ± 5.0
**1**	268.6 ± 31.5	8.5 ± 0.9	6.6 ± 1.9
**2**	113.4 ± 21.8	7.6 ± 0.2	2.4 ± 0.4
**3**	32.2 ± 2.9	4.0 ± 0.3	1.3 ± 0.5
**4**	0.25 ± 0.08	0.20 ± 0.02	0.62 ± 0.27
**D-22 ^[b]^**	0.98 ± 0.31	1.13 ± 0.19	0.09 ± 0.01
**Prazosin ^[b]^**	1.84 ± 0.48	>100	12.6 ± 2.93
**Corticosterone ^[b]^**	21.7 ± 2.44	34.2 ± 6.47	0.29 ± 0.04

^[a]^ IC_50_ values derived from concentration-response curves. IC_50_ values are quoted with their standard deviations (SD). Values are mean ± SD of 3–4 independent experiments performed in triplicates. ^[b]^ IC_50_ values were taken from ref. [33].

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
