# Peer review of "Insight into the Anticancer Activity of Copper(II) 5-Methylenetrimethylammonium-Thiosemicarbazonates and Their Interaction with Organic Cation Transporters"

_biomolecules, 2020, doi:10.3390/biom10091213_

Round 1
Reviewer 1 Report
I have no intention of diminishing the worth of the presented studies, but in my opinion it needs some improvement. My detail comments are given below.
- First of all the paper is too long, the authors should shorten the Introduction, Conclusion and Section 2.8, lines 534-556 presenting only literature data.
- Line 172 – the first table mentioned in the text is Table 4 then in line 233 - Table 1. If the authors wished to indicate that the results of SC-XRD analysis for compounds 2-4 are in Table 4, they should add “see Section 4.5”, which would explain the sequence of table references.
- Line 215 the authors write “Notably, pKa value of the hydrazinic-N2’H moiety could not be determined in the studied pH range”, which needs some explanation why it was impossible.
- The authors should check the caption to Table 1, saying “[b] logKderived = logβ[CuL]+ - pKa(H2L+) for the equilibrium Cu2+ + H2L+ = [CuL]+ + 2H+” . The values calculated according to [b] are in the Table 1 assigned to logKderived [CuLH-1]+ so MLOH species and not as the authors suggest by [b] to ML species.
- Table 1 - why the observed rate constants (kobs) for the redox reaction of complexes with GSH for [H2LH]Cl are given at such a high value of ± 0.022, while for the other derivatives they are much lower (±0.002, ±0.001 and ±0.005)?
- Line 247 – there is Figure 4a, instead of Figure 5
- Line 301 - there is (Figure 4b) instead of (Figure 6b)
- Line 304 – the authors write “see logKderived [CuL]+ values in Table 1”. However Table 1 does not give the values of logKderived[CuL]+ but those of logKderived[CuLH-1]+.
- The number of water molecules in crystals 2 and 4 given in the text is in disagreement with that following from the empirical formula given in Table 4. What is the reason for this difference, is it related to the error in structures refinement?
- Conclusion should be more concise.
Author Response
1) First of all the paper is too long, the authors should shorten the Introduction, Conclusion and Section 2.8, lines 534-556 presenting only literature data.
Response: Taking into account that other two reviewers did not suggest the shortening of the initial text of the manuscript, we reduced the text by removing several sentences in Introduction, Conclusion and Section 2.8. The paragraph (line 534-556) in Section 2.8 is re-worked according to the critical comments of this reviewer.
2) Line 172 – the first table mentioned in the text is Table 4 then in line 233 - Table 1. If the authors wished to indicate that the results of SC-XRD analysis for compounds 2-4 are in Table 4, they should add “see Section 4.5”, which would explain the sequence of table references.
Response: We thank to reviewer for this suggestion. The text was added (Page 5, line 171).
3) Line 215 the authors write “Notably, pKa value of the hydrazinic-N2’H moiety could not be determined in the studied pH range”, which needs some explanation why it was impossible.
Response: However, the studied compounds have two dissociable protons (OH and hydrazinic NH), we could determine only one pKa value for each proligand in the studied pH range (2-11.5) and was attributed to the OH group based on the typical UV-visible spectral changes accompanying the deprotonation. The deprotonation of the other moiety takes place at the more basic pH values (pH > 11.5) where the pH cannot be measured accurately enough because of the alkaline error of the electrode. (The hydrazinic NH has always much higher pKa than OH (see Ref. [35]).) The text was modified accordingly (Page 6, line 214-217).
4) The authors should check the caption to Table 1, saying “[b] logKderived = logβ[CuL]+ - pKa(H2L+) for the equilibrium Cu2+ + H2L+ = [CuL]+ + 2H+” . The values calculated according to [b] are in the Table 1 assigned to logKderived [CuLH-1]+ so MLOH species and not as the authors suggest by [b] to ML species.
Response: The reviewer is right. As the proligands have theoretically 2 pKa values, but only one could be determined in the studied pH range, in the complexation studies Cu2+, H+ and (HL) species were considered as components from which the forming complexes were “built up”. This important information is added now for the definition of the various equilibrium constants in Table 1. As complexes [CuL]+ predominate at the physiological pH, we wanted to compare the stability of this type of complexes. The calculated derived constants belong to the [CuL]+ species according to the calculation now in the legend [d], but the name of the derived constant was wrong in the Table. The Table was modified accordingly (Page 7).
5) Table 1 - why the observed rate constants (kobs) for the redox reaction of complexes with GSH for [H2LH]Cl are given at such a high value of ± 0.022, while for the other derivatives they are much lower (±0.002, ±0.001 and ±0.005)?
Response: We thank the reviewer for this notice. We have re-evaluated our parallel measurements on the Cu(II) complex of [H2LH]Cl, and after negligence of one of the runs, a somewhat lower standard deviation was obtained. Notably, this error is still higher than in the case of the other complexes. The data was modified accordingly (Page 7).
6) Line 247 – there is Figure 4a, instead of Figure 5
Response: We thank the reviewer for this notice. It was corrected.
7) Line 301 - there is (Figure 4b) instead of (Figure 6b)
Response: We thank the reviewer for this observation, it was corrected.
8) Line 304 – the authors write “see logKderived [CuL]+ values in Table 1”. However Table 1 does not give the values of logKderived[CuL]+ but those of logKderived[CuLH-1]+.
Response: It was corrected (see the answer for question 4).
9) The number of water molecules in crystals 2 and 4 given in the text is in disagreement with that following from the empirical formula given in Table 4. What is the reason for this difference, is it related to the error in structures refinement?
Response: There are no errors in structural refinement.
Structure [Cu(HLMe)Cl]Cl·2.5H2O (2). According to structure refinement the exact composition calculated for two asymmetric molecules is C26H49.81Cl4Cu2N8O6.905S2. The fractional values for H and O atoms appear because one of the co-crystallized water molecules is located close to Cl anion, which is disordered in two positions. Consequently, this fragment was modeled assuming the overall occupancy to be equal to 1.0 and the composition can be represented as: C26H40Cl4Cu2N8O2S2·4.905H2O. In the text, the empirical formula was calculated for one complex molecule: C13H20Cl2CuN4OS·2.5H2O. In this case the number of solvate water molecules was round to 2.5 giving rise for disagreement mentioned by referee.
Structure [Cu(HLPh)Cl]Cl·5H2O (4). According to final cif.file the composition is in agreement with the above formula: C18H32Cl2CuN4O6S, as indicated in Table 4.
10) Conclusion should be more concise.
Response: The conclusion was re-worked by removing of some excessive sentences.
Reviewer 2 Report
In the here presented manuscript “Insight into the Anticancer Activity of Copper(II) 5-Methylenetrimethylammonium-Thiosemicarbazonates and Their Interaction with Organic Cation Transporters” authors described a series of newly synthesized water-soluble salicylaldehyde thiosemicarbazones with their copper(II) complexes. This publication describes anticancer properties of the studies compounds with regard to their interaction with organic cation transporters OCT1–3. This approach is innovative and brings fresh insight into the medicinal chemistry of thiosemicarbazones. Presented compounds were broadly characterized by analytical, spectroscopic and X-ray diffraction methods.
I recommend to publish this manuscript in the Biomolecules.
Author Response
Response: We thank to this reviewer for the revision our manuscript.
Reviewer 3 Report
The reviewed manuscript describes the synthesis, antiproliferative activity and the interaction with Organic Cation Transporters (OCT 1-3) of the four new water soluble methylenetrimethylammonium thiosemicarbazones and their Cu(II) complexes. Some of the obtained compounds show an interesting antiproliferative activity comparable or even higher than cisplatin. Furthermore, it turned out that the Cu(II) complexes of the title thiosemicarbazones have shown the interaction with OCT1-3 comparable with control inhibitors. Additionally, authors determine the stability of obtained complexes in DMSO/water solutions, as well as proton dissociation, stability constants, and distribution coefficients. It is worth pointing out that three of the prepared complexes have been characterized by X-ray diffraction analysis.
Overall, in my opinion the reviewed manuscript presents very interesting, multidisciplinary results, especially in the field of biological properties of thiosemicarbazones and its complexes. Therefore, I recommend the publication of reviewed manuscript in Biomolecules after some revisions according to the following comments.
- On page 4 (lines 149-51) Authors state that “The starting aldehyde salt was obtained by the reaction of 5-chloromethylsalicylaldehyde with trimethylamine in tetrahydrofuran (THF) by following a literature protocol[32, 33]”, however ref. 32 and 33 describe preparation of the
5-chloromethylsalicylaldehyde, not mentioned aldehyde salt. Such information may mislead the potential readers, thus above sentence and similar information in chapter 4.1 (page 18, lines 683-684) have to be corrected. Moreover, the synthetic procedure for chloride salt of
5-(methylenetrimethylammonium) salicylaldehyde together with spectroscopic data (at least 1H and 13C NMR) have to be added to the experimental section.
- Page 4, line 148: it should be “5-methylenetrimethylammonium” not
“5-methylenetrimethylamonium”.
- I cannot agree with the statement (page 5, lines 192-193) that “The performed characterisation (1H and 13C NMR, IR, ESI-MS, SC-XRD) of the prepared proligands and complexes provided evidence for the purity”. NMR, IR, ESI-MS and SC-XRD characterization confirm the structure of the obtained compounds, but not their purity. The purity could be determined by HPLC, qNMR or elemental analysis (as it was done by Authors).
-Page 19, lines 724-725: multiplicity of the signal at 3.62 ppm of methylene group (C13H2) should be checked. It should be quartet or doublet of quartets, but not doublet of doublets.
- Page 19, line 732-733: the signal at 7.64 ppm (C14-H and C18-H) looks just like doublet (1H NMR spectrum at page 11 in SI) not multiplet (m).
-Page 19, line 733-734: multiplicity of the signal at 7.11 ppm of C3-H should be checked. In my opinion it is a doublet of AB spin pattern with characteristic “roof” effect. The third line which at first glance looks like the side line of triplet is just a broad signal of traces of some impurity which partially overlap with the doublet (please look at the zoom of this signal on the HSQC spectrum at page 12 in SI)
Author Response
1) On page 4 (lines 149-51) Authors state that “The starting aldehyde salt was obtained by the reaction of 5-chloromethylsalicylaldehyde with trimethylamine in tetrahydrofuran (THF) by following a literature protocol[32, 33]”, however ref. 32 and 33 describe preparation of the
5-chloromethylsalicylaldehyde, not mentioned aldehyde salt. Such information may mislead the potential readers, thus above sentence and similar information in chapter 4.1 (page 18, lines 683-684) have to be corrected.
Response: The synthesis of the chloride salt of 5-(methylenetrimethylammonium) salicylaldehyde was described in the work of Tanaka T. et al, Bull. Chem. Soc. Jpn. 1997, 70, 615-629. We cited this work as reference 32b in the manuscript now (Page 4, line 146-150). The same reference was cited in chapter 4.1 (Page 18, line 678).
2) Moreover, the synthetic procedure for chloride salt of
5-(methylenetrimethylammonium) salicylaldehyde together with spectroscopic data (at least 1H and 13C NMR) have to be added to the experimental section.
Response: The synthetic protocol for chloride salt of 5-(methylenetrimethylammonium) salicylaldehyde was well-established by Tanaka T. et al., Bull. Chem. Soc. Jpn. 1997, 70, 615-629. The compound was spectroscopically characterized via 1H NMR. Therefore, we decided to cite this paper (page 4, line 150) what is in agreement with the author’s guideline for the manuscript preparation.
3) Page 4, line 148: it should be “5-methylenetrimethylammonium” not
“5-methylenetrimethylamonium”.
Response: We thank this reviewer for noting this typo. We corrected the sentence now (page 4, line 147).
4) I cannot agree with the statement (page 5, lines 192-193) that “The performed characterisation (1H and 13C NMR, IR, ESI-MS, SC-XRD) of the prepared proligands and complexes provided evidence for the purity”. NMR, IR, ESI-MS and SC-XRD characterization confirm the structure of the obtained compounds, but not their purity. The purity could be determined by HPLC, qNMR or elemental analysis (as it was done by Authors).
Response: The statement is modified (Page 5, line 191-193).
5) Page 19, lines 724-725: multiplicity of the signal at 3.62 ppm of methylene group (C13H2) should be checked. It should be quartet or doublet of quartets, but not doublet of doublets.
Response: We agree with this statement. The multiplicity of the signal of methylene C13H2 is doublet of quartets as it couples with neiborghing protons of the NH and CH3 group. We corrected it now in the text and in SI (Figure S9).
6) Page 19, line 732-733: the signal at 7.64 ppm (C14-H and C18-H) looks just like doublet (1H NMR spectrum at page 11 in SI) not multiplet (m).
Response: We agree with this statement. We corrected it now in the text and in the SI (Figure S13).
7) Page 19, line 733-734: multiplicity of the signal at 7.11 ppm of C3-H should be checked. In my opinion it is a doublet of AB spin pattern with characteristic “roof” effect. The third line which at first glance looks like the side line of triplet is just a broad signal of traces of some impurity which partially overlap with the doublet (please look at the zoom of this signal on the HSQC spectrum at page 12 in SI).
Response: We agree that the multiplicity of the signal at 7.11 is a doublet. We corrected it now in the text and in SI (Figure S13).